# Vision Transformers Don't Need *Trained* Registers

**Nick Jiang**[*]    **Amil Dravid**[*]    **Alexei A. Efros**    **Yossi Gandelsman**

UC Berkeley
[*]Equal contribution
`{nickj,amil_dravid,aaefros,yossi_gandelsman}@berkeley.edu`

## Abstract

We investigate the mechanism underlying a previously identified phenomenon in Vision Transformers – the emergence of high-norm tokens that lead to noisy attention maps (Darcet et al., 2024). We observe that in multiple models (e.g., CLIP, DINOv2), a sparse set of neurons is responsible for concentrating high-norm activations on outlier tokens, leading to irregular attention patterns and degrading downstream visual processing. While the existing solution for removing these outliers involves retraining models from scratch with additional learned *register tokens*, we use our findings to create a training-free approach to mitigate these artifacts. By shifting the high-norm activations from our discovered *register neurons* into an additional untrained token, we can mimic the effect of register tokens on a model already trained without registers. We demonstrate that our method produces cleaner attention and feature maps, enhances performance over base models across multiple downstream visual tasks, and achieves results comparable to models explicitly trained with register tokens. We then extend test-time registers to off-the-shelf vision-language models, yielding cleaner attention-based, text-to-image attribution. Finally, we outline a simple mathematical model that reflects the observed behavior of register neurons and high norm tokens. Our results suggest that test-time registers effectively take on the role of register tokens at test-time, offering a training-free solution for any pre-trained model released without them.[1]

## 1   Introduction

Vision Transformers (ViTs) (Dosovitskiy et al., 2021) have become a dominant architecture in computer vision, offering strong performance across a wide range of tasks (Oquab et al., 2024; Radford et al., 2021). Recently, Darcet et al. (2024) observed a surprising property in these models: the emergence of high-norm intermediate tokens at seemingly random locations in the image during the internal computation of the ViT (see "Original" column in Figure 1). These outlier tokens were shown to appear in low-information image areas (e.g., uniform background patches) and were demonstrated to capture global image information.

Darcet et al. (2024) interpreted these high-norm tokens as a form of emergent global memory – a mechanism through which the model stores and retrieves global information, analogous to registers in CPUs. Based on this interpretation, they proposed to eliminate these outlier image tokens by augmenting the input with dedicated non-image tokens during training, calling them "register tokens." This allows the patch tokens to focus solely on encoding local content, leading to improved performance on dense visual tasks. However, this technique requires re-training existing models from scratch with these extra register tokens, limiting its applicability in practice.

---

[1]Project Page: `https://avdravid.github.io/test-time-registers`
 Code: `https://github.com/nickjiang2378/test-time-registers`

39th Conference on Neural Information Processing Systems (NeurIPS 2025).

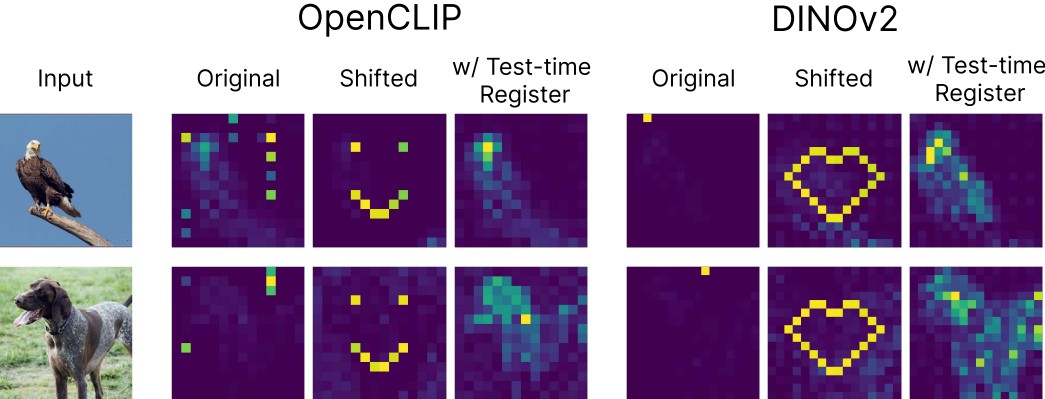

Figure 1: **Controlling high-norm tokens in Vision Transformers.** As shown in Darcet et al. (2024), high-norm outlier tokens emerge in ViTs and lead to noisy attention maps ("Original"). By identifying the mechanism responsible for their emergence, we demonstrate that we can shift these outlier tokens to arbitrary positions at test time ("Shifted"). Shifting the outlier tokens *outside* of the image mimics register behavior at test-time ("w/ Test-time Register"), resulting in more interpretable attention patterns and downstream performance comparable to models retrained with registers.

In this work, we argue that while registers are indeed useful, models don't need to be retrained with them. Instead, we show that registers can be added *post hoc*, without any additional training. To do this, we first investigate the mechanism underlying the emergence of high-norm tokens. We identify a sparse set of neurons – *register neurons* – that create the outlier tokens by contributing high-norm values to them. By directly editing the activation maps of the register neurons during the forward pass of the network, we can induce the formation of the high-norm activations at arbitrary token positions (Figure 1). We use it to create new register tokens at *test-time*, even in models that were never trained with them, by appending new tokens and activating the register neurons in their positions.

We show that models with test-time registers provide comparable performance to models with trained registers on various downstream tasks (e.g., classification, segmentation, and depth prediction) and largely improve over models without registers for unsupervised object discovery (20-point correct localization improvement) and attention-based zero-shot segmentation (+5 mIOU). Next, we demonstrate that test-time registers reduce high-norm artifacts in vision-language models, improving the alignment between textual outputs and relevant visual regions. Finally, we present a simple mathematical model capturing the observed behavior of register neurons and high-norm tokens, empirically finding that test-time attention biases can mitigate outliers and attention sinks without *any* additional tokens.

In summary, our contributions are as follows:

- We determine the mechanism behind the creation of high-norm tokens in ViTs by finding *register neurons* that, when activated, cause the appearance of these outliers (Section 3.1). A simple mathematical model captures the observed behavior of these neurons (Section 6).

- We demonstrate that activating the register neurons at test-time in other image locations shifts the high-norms to the corresponding tokens (Section 3.2).

- We present a training-free method for adding registers to models that were trained without them, by appending additional tokens and activating register neurons in their positions (Section 4).

- We evaluate the performance of models with test-time registers and show that it is comparable to models with trained registers, thus eliminating the need for retraining models with registers from scratch (Section 5).

## 2   Related work

**Feature visualization in vision models.** Visualizing features of computer vision models has been used for diagnostics long before the transition of the field to deep-learning (e.g., Vondrick et al. (2013)). Features in early CNN-based models were visualized to interpret their emergent computation (Zeiler

& Fergus, 2014; Bau et al., 2017) and to approximate saliency maps (Itti et al., 2002). In ViTs, the attention map from the [CLS] token has been used for various attribution methods (Caron et al., 2021; Chefer et al., 2021), and has also been steered to improve model performance (Shi et al., 2023). Darcet et al. (2024) showed that the newer ViT-based models (Oquab et al., 2024; Radford et al., 2021) exhibit artifacts in their attentions, affecting their applicability for visualization and downstream tasks. These artifacts were connected to high-norm tokens that emerge in the Transformers.

**High-norm tokens in Transformers.** Transformers tend to create high-norm tokens during their internal computation, when trained on language tasks (Xiao et al., 2024) or on vision tasks (Darcet et al., 2024). For language models, Xiao et al. (2024) showed that Transformers allocate excessive attention to these high-norm tokens, and named them "attention sinks." Sun et al. (2024) demonstrated that "attention sinks" emerge due to previous massive activations in the residual stream. Yona et al. (2025) linked the emergence of "attention sinks" to the inability of language models to repeatedly generate a single token, and suggested a test-time fix by zeroing out the relevant activated neurons. For vision models, Darcet et al. (2024) demonstrated the emergence of high-norm tokens in low-informative areas and suggested retraining the model with extra tokens ("registers") to remove these large norms from the image patches. Wang et al. (2024) used a simpler fine-tuning approach in DINOv2 to avoid complete re-training. Nakamura et al. (2024) suppressed artifacts during inference by modifying last-layer attention, showing that it improved image clustering performance. In contrast, we remove high-norm outliers *at their source* by editing the activations of the neurons that create them, aiming to eliminate their effect on later computations.

**Explaining neural functionality in vision models.** The role of individual neurons (post non-linearity single channel activations) has been broadly studied in vision models, demonstrating that some neurons recognize low-level image features as well as high-level perceptual and semantic properties of the inputs (Schwettmann et al., 2023; Bau et al., 2017; Hernandez et al., 2021; Gandelsman et al., 2025; Dravid et al., 2023). Nevertheless, most of the research has focused on linking neural behavior to features of the input or output, neglecting other possible neural functionality that may not be related to any specific image feature. Robinson et al. (2024) discovered sparse neural activations that indicate the absence of features rather than serving as feature detectors. Sun et al. (2024) found massive activations in ViTs that operate as constant biases for attention layers. Differently, we discover and edit neuron activations responsible for creating high-norm tokens in ViTs to mimic registers.

## 3 Interpreting the outlier tokens mechanism in ViTs

As shown by Darcet et al. (2024), high-norm outlier patches emerge during inference in various pre-trained ViTs. These patches strongly draw attention from the [CLS] token, resulting in artifacts in the attention maps (Figure 1). Outlier patches appear primarily in areas that exhibit high similarity to their neighboring areas (e.g., uniform background patches). Moreover, they were shown to capture global image information and lose their local patch contents (pixel + position). In this section, we study how the outlier tokens emerge in ViTs during the forward pass and discover a sparse set of neurons whose activations dictate the location of outliers. We use the term "neuron" to denote a single hidden unit in the MLP layers of a Transformer block, i.e., the scalar output after the linear transformation and nonlinearity. We then show that we can edit these neurons to move the outlier positions. Our main analysis is applied to OpenCLIP ViT-B/16 (Cherti et al., 2023), with similar findings on DINOv2-L/14 (Oquab et al., 2024) shown in Section A.12.

### 3.1 How do outliers emerge in ViTs?

**Outlier patches appear after MLPs.** To identify the Transformer component most responsible for outlier patches, we track the maximum norm of image patches after attention blocks and MLPs across 1000 ImageNet images (Deng et al., 2009). Figure 2 shows that outliers appear after the MLP of layer 6 in OpenCLIP. We also measure the maximum weight the [CLS] token attends to any patch across all heads in each layer and observe that high attention occurs after the layer 6 MLP. This observation suggests that the MLP increases the norms of certain patches, consequently creating attention sinks.

**A small, consistent set of neurons activate highly on outlier positions.** Examining layer 6, we find that just five MLP neurons exhibit consistently high contributions at the top outlier patch across images (see Section A.11.1). When inspecting the activation maps of three such neurons (Figure 3, more in Section A.10), we observe that they sparsely activate on all outlier positions, not just the top

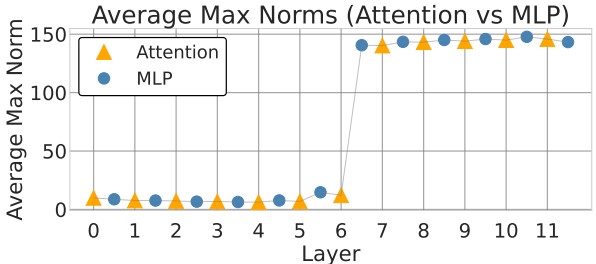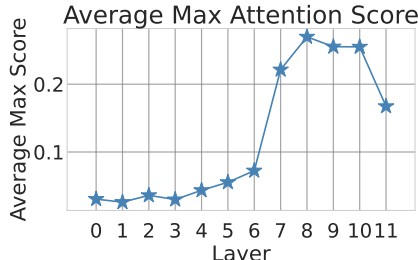

Figure 2: **Outlier patches appear after MLPs; attention sinks appear after outlier patches.** Left: Max norms across image patches (OpenCLIP ViT-B/16). Right: max attention scores of the [CLS] token in the last layer. In both plots, we average across 1000 images. The outlier norms and attention sinks occur in consecutive layers.



Figure 3: **Highly activated neurons on the top outlier activate on all outlier positions.** We present activation maps of three neurons from layer 6 that activate highly on the top outlier patch. These maps near-perfectly align with the high-norm outliers ("Patch Norms").

outlier position. This suggests that these high-activating neurons are not position-specific, but rather, responsible for outliers generally. Given these observations, we develop a method to automatically detect these neurons in the next section.

## 3.2 Register neurons

Based on our previous analysis, we hypothesize that a small, consistent set of sparsely activating neurons control where outliers emerge. These neurons appear in the preceding layers before outliers form. Given their importance for the formation of outliers, we call them "register neurons."

**Detecting register neurons.** Based on our hypothesis, we formulate a simple algorithm to find register neurons in Algorithm 1. Our method finds neurons whose average activation at outlier positions is consistently high across images. To detect outlier positions (FINDOUTLIERS within Algorithm 1), we follow Darcet et al. (2024) and output the positions for which the norms of the corresponding image tokens are above a predefined threshold. Our algorithm searches for neurons in preceding layers before outliers start, set by the $top\_layer$ parameter. We output $top\_k$ neurons with the highest average activations. We present a discussion of hyperparameters in Section A.4. For OpenCLIP, we set $top\_layer = 5$, the outlier threshold at 75, and $top\_k = 10$. Since register neurons determine where outliers emerge, we can also intervene upon them to move outliers to arbitrary positions, as demonstrated next.

---

**Algorithm 1** FINDREGISTERNEURONS

1: **Input:** Image set $\mathcal{I} = \{I_1, \ldots, I_M\}$, maximum layer index $top\_layer$, number of register neurons to return $top\_k$, number of neurons in each layer $N$
2: **Output:** $top\_k$ register neurons
3: $avg\_act \leftarrow \mathbf{0}_{top\_layer \times N}$ # initialize array for average activations
4: **for all** $I_i \in \mathcal{I}$ **do**
5:    $O \leftarrow$ FindOutliers($I_i$) # get indices of top-norm patches
6:    **for** $\ell = 0$ to $top\_layer$ **do**
7:       **for** $n = 0$ to $N - 1$ **do**
8:          $avg\_act[\ell, n] \leftarrow avg\_act[\ell, n] + \frac{1}{|O|M} \sum_{p \in O} \text{activation}_{\ell, n}(I_i, p)$
9:       **end for**
10:    **end for**
11: **end for**
12: **return** $top\_k$ neurons with largest $avg\_act$

---

**Moving outliers with register neurons.** To demonstrate the importance of these register neurons, we can use them to "move" outliers to different image locations. We first apply FINDREGISTERNEURONS to detect the register neurons. We then modify their activation pattern during the forward pass – for each register neuron, we copy the highest neuron activation across the tokens into the locations of the tokens to which we want to move the outliers. We zero out the activations of the neuron elsewhere.

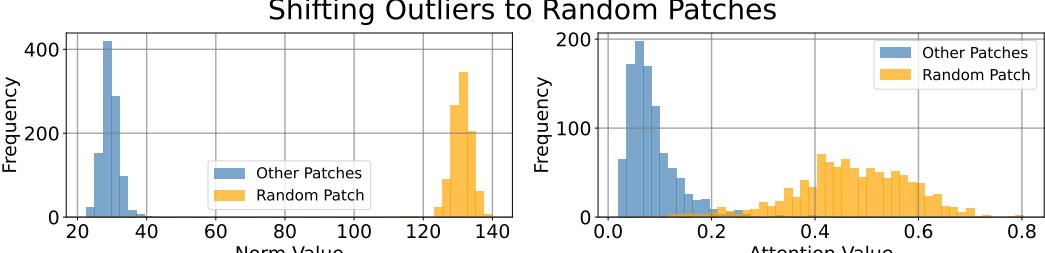

Figure 4: **Intervening on activations of register neurons effectively shifts outliers to random patches and test-time registers.** For all register neurons, we copy their highest activation into a selected patch and zero out the activations elsewhere. Left: norm of chosen random patch (yellow) and max norm of any other patch (blue). Right: `[CLS]` attention to chosen random patch (yellow) and max `[CLS` attention (blue) to any other patch. Our intervention can shift outliers to randomly selected patches as well as test-time registers (see Section A.11.2).

**Register neurons causally set the position of outliers.** To test our ability to move outliers to arbitrary spots, we assign the highest activation to a random patch and measure its last-layer output norm, the maximum norm of other patches, and the highest last-layer `[CLS]` attention, both to the selected outlier and any other patch. Successful interventions yield high norms and attention for the targeted patch, with low values elsewhere. As shown in Figure 4, modifying the activations of register neurons effectively controls where outliers emerge. In contrast, intervening on random neurons has little effect (Section A.11.3). Figure 1 demonstrates that intervening on register neurons can make outliers appear in various counts and spatial patterns (e.g. a heart). We use this technique to mimic registers, as follows in the next section.

## 4 Adding registers at test-time

Given that register neurons can be used to move outliers arbitrarily, we investigate moving outliers outside the image entirely into extra tokens we call "test-time registers."

**Moving outliers to added tokens.** As outlier patches lose their local patch information (Darcet et al., 2024), it is undesirable to have them within the image since it may affect downstream performance. Previous work has suggested retraining ViTs with extra tokens to remove high-norm artifacts and attention sinks. However, retraining existing ViTs is expensive, so we propose to add an extra input token and move the outliers there with register neurons. Algorithm 2 (Section A.3) with accompanying visualization in Figure 11, specifies our test-time register edit. At each register neuron, we copy the maximum patch activation to the register token, set all other token activations to zero, and resume the forward computation. Importantly, these outliers are essential to the model's internal computations and must appear within a token. For instance, zeroing out the register neurons' activation maps instead of moving the outliers causes OpenCLIP ViT-B/16's zero-shot ImageNet performance to drop ∼15%. Ablating a set of random neurons results in minimal drop: $69.3\% \pm 1.1$.

**Test-time register initialization.** We initialize our extra token to be a vector of zeros. We assess several initialization strategies, but we find that they do not significantly affect the ability of test-time registers to store the high norms (Section A.6). Additionally, we focus our investigation on using one test-time register and report the impact of using more registers in Section A.5.

**Outliers move outside the image after adding test-time registers.** To evaluate whether test-time registers can absorb the outliers, we apply our intervention and measure the max norms of image patch outputs and the test-time register. Our intervention results are nearly identical to the distribution of patch norms and `[CLS]` attention after shifting outliers to random patches, previously shown in Figure 4 (see results for test-time registers in Section A.11.2). The image patches no longer have outliers, whereas the test-time register absorbs the outlier norms. This change is also present in the attention maps (Figure 5), which no longer have noisy artifacts and match the quality of attention maps from models with trained registers.[2] See Section A.9 for attention maps in all model layers.

---

[2]As there is no open-source OpenCLIP trained with registers, we only present comparisons on DINOv2.

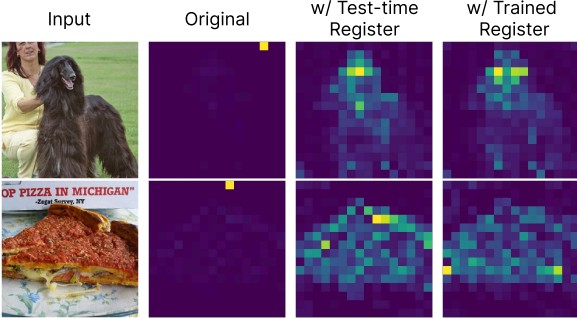

| | IN1k | CF10 | CF100 |
|---|---|---|---|
| [CLS] token | 85.6 | 99.4 | 93.4 |
| central token | 73.3 | 98.0 | 88.1 |
| outlier token | 84.5 | 99.2 | 92.8 |
| trained register | 83.1 | 99.2 | 93.0 |
| test-time register | 84.5 | 99.1 | 93.0 |

Figure 5: **Qualitative results on attention maps w/ test-time registers.** We present the last layer's mean [CLS] attention maps in DINOv2 and compare them to the model with trained registers. Test-time registers produce similarly high-quality maps as trained registers.

Table 1: **Linear probing classification results (DINOv2 ViT-L/14).** Test-time registers achieve higher performance on linear probing than non-outlier tokens, suggesting that they hold global information similarly to trained registers. They match the performance of outlier tokens, indicating that they have absorbed the role of outliers.

**Test-time registers hold global information.** We verify that test-time registers encapsulate global image information (e.g., image class) similarly to trained registers (Darcet et al., 2024). To assess this, we perform linear probing on both trained and test-time registers for classification on ImageNet (Deng et al., 2009), CIFAR-10, and CIFAR-100 (Krizhevsky et al., 2009). We also compare their performance to the [CLS] token and a token that corresponds to a central patch in the image. As shown in Table 1, the classification accuracy of test-time registers closely matches that of trained registers and is slightly lower than the [CLS] token performance. These results suggest that test-time registers, like their learned counterparts, effectively capture global image-level information.

## 5 Experiments

We evaluate how adding test-time registers affects the downstream performance of models trained without registers. We begin by detailing the evaluated models, then compare performance across classification, dense prediction, unsupervised object discovery, and zero-shot segmentation tasks, finding that test-time registers perform comparably to their retrained counterparts. Finally, we apply test-time registers to an off-the-shelf vision-language model to improve its interpretability. We present additional experiments on preventing typographic attacks in Section A.8.

**Models.** We evaluate using DINOv2 (Oquab et al., 2024) and OpenCLIP (Cherti et al., 2023). For DINOv2, we use the publicly released ViT-L/14 checkpoints trained on LVD-142M, including both the standard model and a variant trained with four registers. These two models serve as our baselines, while our approach applies edits to the standard model. For OpenCLIP, we evaluate the ViT-L/14 and ViT-B/16 models trained on LAION-2B (Schuhmann et al., 2022). As no checkpoints with trained registers are available for OpenCLIP, we only compare our approach to the standard models. We present results on larger models in Section A.7.

### 5.1 Classification and dense prediction

**Linear probing.** We conduct linear probing on ImageNet classification (Deng et al., 2009), ADE20k segmentation (Zhou et al., 2017), and NYUv2 monocular depth estimation (Nathan Silberman & Fergus, 2012), following the procedure outlined in (Oquab et al., 2024; Darcet et al., 2024).

The results in Table 2 show that models edited with test-time registers maintain their performance on ImageNet classification with slight performance gains over the base model on segmentation and depth estimation tasks. These improvements are consistent with the performance gains observed in DINOv2 explicitly trained with registers. Thus, we demonstrate that intervening on a model with test-time registers does not degrade the model's representations, and in some cases, even enhances performance on prediction tasks. While we observe a small improvement in classification performance with the DINOv2 model trained with registers, we note that this was an independently trained model. Hence, the difference could be due to variations in initialization and training dynamics, rather than the effect of trained registers fixing artifacts. We report $95\%$ confidence intervals for these results in Section A.13 and find that performance differences remain consistent.

| | IN Top-1 ↑ | ADE20k mIoU ↑ | NYUd rmse ↓ |
|---|---|---|---|
| DINOv2 ViT-L/14 | 86.4 | 48.3 | 0.388 |
| w/ trained registers | 86.7 | 49.1 | 0.382 |
| w/ test-time register | 86.4 | 49.1 | 0.378 |
| OpenCLIP ViT-B/16 | 77.4 | 40.1 | 0.603 |
| w/ test-time register | 77.5 | 40.3 | 0.596 |

Table 2: **Linear probing results.** The performance of models with test-time registers maintains or improves performance over the original models, and largely matches models with trained registers.

| OpenCLIP | Acc. |
|---|---|
| ViT-L/14 | 76.4 |
| w/ test-time register | 76.4 |
| ViT-B/16 | 71.3 |
| w/ test-time register | 71.3 |

Table 3: **OpenCLIP zero-shot ImageNet classification.** Adding test-time registers maintains performance.

**Zero-shot classification.** We evaluate zero-shot ImageNet classification with OpenCLIP to assess whether test-time registers preserve the semantic structure of the original representation space. Unlike linear probing, where a trained linear head can compensate for small shifts in representation, zero-shot classification is more sensitive to such changes. We compare zero-shot performance before and after applying test-time registers across both ViT-L/14 and ViT-B/16 in Table 3, observing that the intervention does not sacrifice performance.

## 5.2 Zero-shot segmentation

To validate that test-time registers result in more interpretable attention maps, we follow a standard protocol for evaluating heatmap-based explainability methods (Hooker et al., 2019) – binarizing the heatmap into a foreground/background segmentation map, and evaluating its segmentation quality. We compute the mean attention map for the [CLS] token in the last layer (Chefer et al., 2021) for the original model without registers, the model with test-time registers, and a model trained with registers if available. We evaluate zero-shot segmentation performance on ImageNet-segmentation (Guillaumin et al., 2014), which contains 4,276 images from the ImageNet validation set with annotated segmentations.

**Results.** We present our zero-shot segmentation scores in Table 4. We also qualitatively compare the attention maps for DINOv2 with test-time and trained registers in Figure 5, demonstrating similarly high-quality attention maps. Using test-time registers on both DINOv2 and OpenCLIP outperforms the original model on mean IOU and mAP with minimal drops in pixel accuracy. Test-time registers also show slight boosts over the retrained DINOv2 with registers on mean IOU and mAP, suggesting that test-time registers lead to attention maps as clean as those from trained registers.

## 5.3 Unsupervised object discovery

We evaluate models edited with test-time registers for unsupervised object discovery, extracting their features for downstream processing. Darcet et al. (2024) found that the performance on this task correlates with the smoothness of a model's attention maps, particularly for DINOv2, and showed that attention features from models with trained registers lead to better object localization.

**Evaluation setting.** We apply the LOST (Siméoni et al., 2021) object discovery method on the PASCAL VOC 2007, PASCAL VOC 2012 (Everingham et al., 2010), and COCO 20k (Lin et al., 2014) datasets. Our evaluation involves sweeping over the key, query, or value features over the last four layers, and manually adding a bias value to the Gram matrix of features as suggested by Darcet et al. (2024). We compare DINOv2 and OpenCLIP edited with test-time registers against the unedited models and, if available, those with trained registers.

**Test-time registers improve unsupervised object discovery.** We report correct localization (corloc) scores in Table 5, using the best result across key, query, and value features from the final four layers. We find that LOST performance improves significantly over the base model on features computed with our method for DINOv2 (increase of ~21 corloc). Adding the test-time register closes the gap between the baseline model and a model with trained registers, reaching within ~0−2 corloc. This suggests that the test-time register mimics the role of trained registers on this task. However, we note that test-time registers only marginally affect the results for OpenCLIP, a phenomenon also observed in Darcet et al. (2024) with trained registers. Further analysis can be found in Section A.14.

|  | mIoU | Pix. Acc. | mAP |
|---|---|---|---|
| DINOv2 ViT-L/14 | 38.3 | 87.6 | 80.6 |
| w/ trained registers | 33.9 | 91.4 | 79.0 |
| w/ test-time register | 38.9 | 87.6 | 81.1 |
| OpenCLIP ViT-B/14 | 34.7 | 76.0 | 79.2 |
| w/ test-time register | 40.0 | 74.2 | 82.1 |

Table 4: **Zero-shot segmentation results on ImageNet.** We use the last layer's mean `[CLS]` attention maps and find that test-time registers outperform the original models and DINOv2 models with trained registers on mean IOU and mAP.

|  | VOC07 | VOC12 | COCO |
|---|---|---|---|
| DINOv2 ViT-L/14 | 32.2 | 36.8 | 25.4 |
| w/ trained registers | 56.2 | 60.2 | 42.3 |
| w/ test-time register | 53.8 | 57.9 | 41.9 |
| OpenCLIP ViT-B/16 | 30.8 | 35.9 | 23.4 |
| w/ test-time register | 30.9 | 35.9 | 23.6 |

Table 5: **Unsupervised Object Discovery with LOST** (Siméoni et al., 2021). Adding test-time registers significantly boosts performance for DINOv2, effectively closing the gap with DINOv2 trained with registers.

## 5.4 Applying test-time registers to vision-language models

Beyond discriminative vision models, we explore the effect of test-time registers on vision-language modeling (VLM) tasks. Adding a test-time register to the image encoder of a VLM preserves performance on several multimodal benchmarks while improving the interpretability of feature maps.

**Evaluation setting.** Using Algorithm 1, we find a set of 100 register neurons (out of ∼100K neurons) from CLIP ViT-L/14 vision encoder of LLaVA-Llama-3-8B.[3] We then create a test-time register to collect their activations and evaluate the model on the eight main benchmarks from the VLMEvalKit toolkit (Duan et al., 2024). The benchmarks span across OCR, chart interpretation, visual Q/A, etc.

**Test-time registers maintain performance.** We report our benchmark results in Table 6. Overall, we observe that adding a test-time register preserves performance for multi-modal processing. We do not pass the register to the LLM; we leave exploring its use as global memory for the LLM or leveraging multiple registers for adaptive computation to future work.

**Test-time registers improve VLM interpretability.** While adding a test-time register has only a minor impact on performance, it improves the interpretability of cross-modal attention maps, as illustrated in Figure 6. We visualize the norms of the patch outputs from the vision encoder, highlighting the presence of outlier tokens. Next, we visualize the average attention – aggregated across all layers and heads of the language model –

| Benchmark | Baseline | w/ Test-time Register |
|---|---|---|
| Avg. | 46.2 | 46.2 |
| HallusionBench | 28.6 | 29.4 |
| MMVet | 33.4 | 33.9 |
| MMMU_VAL | 40.4 | 40.1 |
| OCRBench | 41.6 | 41.3 |
| MMStar | 46.3 | 46.4 |
| MathVista | 40.9 | 41.3 |
| AI2D | 69.9 | 69.4 |
| MMBenchv1.1 | 68.5 | 68.0 |

Table 6: **Adding a test-time register maintains overall performance across multi-modal tasks.**

from the token responsible for answering the question to the visual tokens. Without registers, the outlier visual tokens create artifacts in the language model's attention. However, adding a test-time register removes outliers and results in more interpretable text-to-vision attribution. This provides clearer insights into the model's behavior (e.g., the attention map shows incorrect localization of the "man closest to us"). We provide more visualizations in Section A.15.

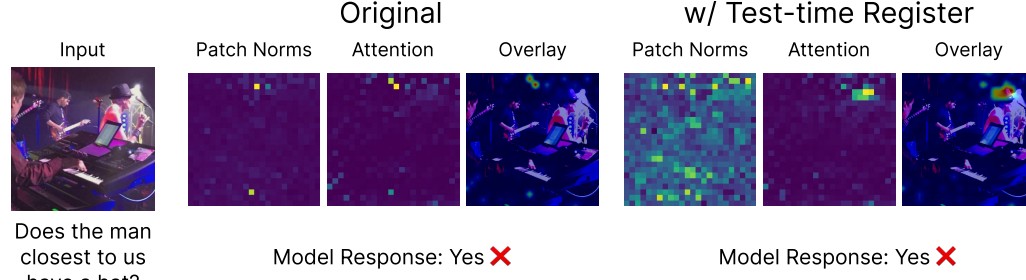

Figure 6: **Test-time registers improve interpretability of LLaVA-Llama-3-8B.** We visualize the patch norms of the vision encoder before projection and the average attention from the answer token to the visual tokens. We observe that outliers leak into the language model's attention to visual tokens, while adding a test-time register mitigates this and leads to more interpretable maps.

---

[3] https://huggingface.co/xtuner/llava-llama-3-8b

# 6 A mathematical model for register neurons

In this section, we present a simple mathematical model describing how register neurons can induce high-norm tokens that attract excess attention. We then relate this behavior to our empirical findings. Prior work has examined the emergence of outlier tokens in large language models and proposed several explanations, most notably the *no-op* (no-operation) hypothesis (Clark et al., 2019; Kobayashi et al., 2020; Bondarenko et al., 2023), which attributes these effects to the softmax constraint enforcing that attention weights sum to one. As a result, the attention operation forces aggregating *some* information from other tokens, even when current embeddings already contain sufficient information for the task. This constraint can direct excess attention towards low-information tokens, creating attention sinks (Xiao et al., 2024). We show that, under certain conditions, register neurons can produce high-norm tokens that give rise to such attention sinks within the no-op framework.

## 6.1 Analytical model

**Set-Up.** We consider the MLP layer from the penultimate Transformer block followed by the full final block (attention + MLP) of a Vision Transformer without normalization layers. Let the input sequence to the penultimate MLP be a task-specific [CLS] token and $n$ patch tokens: $x_{cls}^{(0)}, x_1^{(0)}, ... x_n^{(0)} \in \mathbb{R}^d$ The MLPs will consist of two linear layers (no bias) and an activation function $\phi$ (e.g., ReLU or GELU). We denote the weight matrices of the penultimate block's MLP as $W_{in}^{(1)} \in \mathbb{R}^{d \times d_{mlp}}, W_{out}^{(1)} \in \mathbb{R}^{d_{mlp} \times d}$, and those of the final block's MLP as $W_{in}^{(2)} \in \mathbb{R}^{d \times d_{mlp}}, W_{out}^{(2)} \in \mathbb{R}^{d_{mlp} \times d}$. There will be one attention head with identity projection matrices: $W_Q = W_K = W_V = W_O = I_d \in \mathbb{R}^{d \times d}$. Finally, the output task-specific head is a linear projection that uses the [CLS] token's embedding for prediction: $W_{head} \in \mathbb{R}^{d \times c}$, where $c$ is the number of output classes (or regression targets). Our analysis is carried out in the regime where both $W_{in}^{(2)}$, $W_{head}$ are low-rank, i.e., $\text{rank}(W_{in}^{(2)}) < \min(d, d_{mlp})$ and $\text{rank}(W_{head}) < \text{rank}(d, c)$. Hence, both matrices possess non-trivial kernels.

**Proposition 1 (Register neuron induces attention sink and no-op attention)** *Let $u_1 = (W_{in}^{(1)})_{:,1}$ be a register neuron and $u_2 = (W_{out}^{(1)})_{1,:}$ be the corresponding row in the MLP's second weight matrix, with $\|u_2\| \gg \|(W_{out}^{(1)})_{j,:}\|$ for $j \neq 1$. If both $u_1, u_2 \in \ker(W_{in}^{(2)\top}) \cap \ker(W_{head}^\top)$ and a token $x_s$ is aligned with $u_1$, i.e., $x_s^{(0)} = \beta u_1$ for some $\beta > 0$, then $x_s$ becomes an attention sink and the attention layer has no contribution to the model's output.*

*Proof.* See Section A.1.

**Corollary 1 (Register neuron induces implicit attention bias)** *The contribution of register neurons is equivalent to the attention mechanism with explicit bias terms $(\mathbf{k}', \mathbf{v}')$ as in Sun et al. (2024):*

$$Attention(Q, K, V; \mathbf{k}', \mathbf{v}') = softmax\left(Q \begin{bmatrix} K^T & \mathbf{k}' \end{bmatrix}\right) \begin{bmatrix} V \\ \mathbf{v}'^T \end{bmatrix}$$

*Proof.* See Section A.1.

## 6.2 Empirical validation on OpenCLIP

This mathematical model captures three distinct observed behaviors of register neurons and high norm tokens. We validate this correspondence qualitatively and quantitatively. Notably, our framework admits an equivalent attention formulation augmented with explicit bias terms – values that can be derived directly from register neurons without any additional training.

**(1) Task-Irrelevant Attention Sinks:** The token that achieves the attention sink behavior in our mathematical model lives in the kernel, or null space, of the task specific head, indicating it is not informative of the task. This aligns with the observation that register neurons fire on seemingly random, uninformative regions such as background or uniform texture (e.g., see Figure 3). In the case of language models, attention sinks have been observed to appear in semantically low-information tokens such as <BOS> or delimiters (Xiao et al., 2024).

**(2) Lack of Sink Ruins Performance:** Our framework suggests that zeroing out the activation of a register neuron would disrupt the "no-op" attention behavior, thus unnecessarily altering the [CLS]

token from its intended representation and substantially degrading model performance. To test this, we zero-out the 10 register neurons in OpenCLIP ViT-B/16 and observe a drop in IN1k zero-shot performance from 70.4% to 55.6%. As a baseline, zeroing out a random set of 10 neurons over three trials results in minimal change: $69.3\% \pm 1.1$.

**(3) Register Neurons Induce an Implicit Attention Bias:** Corollary 1 aligns with prior observations (Sun et al., 2024; Gu et al., 2024) that high-norm tokens induce an implicit bias term in the attention mechanism. Sun et al. (2024) proposed learning biases in the attention layers of large language models to remove outliers. Specifically, given the query, key, and value features from $T$ tokens $Q, K, V \in \mathbb{R}^{T \times d}$ for each attention head, they train additional parameters $\mathbf{k}', \mathbf{v}' \in \mathbb{R}^d$ in:

$$\text{Attention}(Q, K, V; \mathbf{k}', \mathbf{v}') = \text{softmax}\left(\frac{Q\left[K^T \quad \mathbf{k}'\right]}{\sqrt{d}}\right)\begin{bmatrix} V \\ \mathbf{v}'^T \end{bmatrix} \tag{1}$$

To test that register neurons implicitly induce attention biases as predicted by Corollary 1, we use OpenCLIP ViT-B/16. For each attention head, $\mathbf{k}'$ and $\mathbf{v}'$ are set to the mean key and value vectors of the test-time register token, which captures the aggregate contribution of the register neurons, averaged across 1000 images. Instead of a test-time register, we zero out the register neurons and inject the constant vectors $\mathbf{k}'$ and $\mathbf{v}'$ directly into the attention computation as above at interference. This maintains the 71.3% IN1k zero-shot performance while suppressing artifacts in attention maps (Figure 7) and mitigating outliers completely from the residual stream (details in Section A.2).

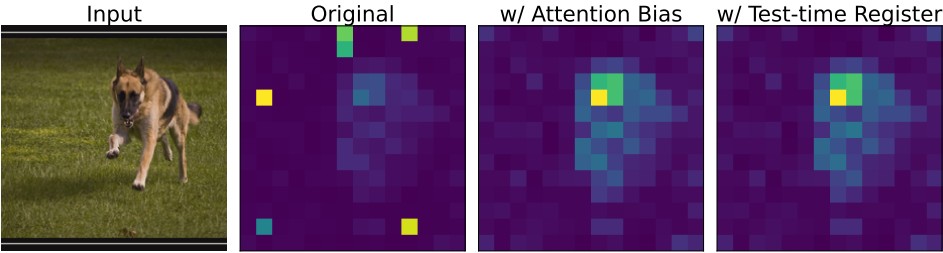

Figure 7: **Register-based attention bias mitigates artifacts.** Mean [CLS] attention maps from the last layer of OpenCLIP ViT-B/16 show that attention bias terms derived from test-time registers (Equation (1)) suppress artifacts, matching the effect of explicitly using test-time registers.

## 7 Discussion, limitations, and future work

We uncovered a simple emergent mechanism in ViTs, a sparse set of neurons that is responsible for creating high-norm tokens in low-information image locations. Editing this mechanism at test-time allowed us to shift the high norms into additional registers, removing artifacts from patches and yielding more interpretable feature maps – while preserving or modestly improving downstream performance. Next, we discuss limitations of our analysis and conclude with future work.

While our analysis shows that we can steer the location of high-norm tokens, we only addressed one component type that is responsible for their creation – neurons, while neglecting other possible elements that can contribute to their formation, such as attention layers or positional encodings (Yang et al., 2024). The edited tokens result in slight performance differences with their learned counterpart (Table 1), suggesting that the test-time registers are not fully equivalent to the high-norm tokens. Finally, similarly to Darcet et al. (2024), we mostly focus on individual models (CLIP, DINOv2). We do not present results on other, less commonly used, pretrained ViTs, and leave it for future work.

The mechanism that we found points to an intriguing property about model neurons – not all neurons have a feature-related role. Register neurons, for example, are responsible for igniting high-norm tokens – an image-independent role – and cannot be discovered by correlating their activations to the image features. Similar high-norm tokens have also been observed in language models (Xiao et al., 2024; Dettmers et al., 2022), suggesting that large language models may rely on a related mechanism. Uncovering other similar input-independent roles of neurons can shed additional light on the computational process of deep neural networks. We plan to develop a methodology for such automatic discovery in future work.

# 8 Acknowledgments

We thank Katie Luo, Lisa Dunlap, Yaniv Nikankin, and Arjun Patrawala for their feedback on our paper. AD is supported by the US Department of Energy Computational Science Graduate Fellowship. YG is supported by the Google Fellowship. Additional support came from the ONR MURI grant and NSF IIS-2403305.

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

# A  Appendix

## A.1  Proof of Proposition 1 and Corollary 1

**Set-Up.** We consider the MLP layer from the penultimate Transformer block followed by the full final block (attention + MLP) of a residual Vision Transformer without normalization layers. Let the input sequence to the penultimate MLP be a task-specific [CLS] token and $n$ patch tokens: $x_{cls}^{(0)}, x_1^{(0)}, ...x_n^{(0)} \in \mathbb{R}^d$ The MLPs will consist of two linear layers (no bias) and an activation function $\phi$ (e.g., ReLU or GELU). We denote the weight matrices of the penultimate block's MLP as $W_{\text{in}}^{(1)} \in \mathbb{R}^{d \times d_{\text{mlp}}}, W_{\text{out}}^{(1)} \in \mathbb{R}^{d_{\text{mlp}} \times d}$, and those of the final block's MLP as $W_{\text{in}}^{(2)} \in \mathbb{R}^{d \times d_{\text{mlp}}}, W_{\text{out}}^{(2)} \in \mathbb{R}^{d_{\text{mlp}} \times d}$. There will be one attention head with identity projection matrices: $W_Q = W_K = W_V = W_O = I_d \in \mathbb{R}^{d \times d}$. Finally, the output task-specific head is a linear projection that uses the [CLS] token's embedding for prediction: $W_{\text{head}} \in \mathbb{R}^{d \times c}$, where $c$ is the number of output classes (or regression targets).

Our analysis is carried out in the regime where both $W_{\text{in}}^{(2)}, W_{\text{head}}$ are low-rank, i.e., $\text{rank}(W_{\text{in}}^{(2)}) < \min(d, d_{\text{mlp}})$ and $\text{rank}(W_{\text{head}}) < \text{rank}(d, c)$. Hence, both matrices possess non-trivial kernels. This low-rank property is consistent with the effective structure observed in practice (Li et al., 2018; Aghajanyan et al., 2021; Hu et al., 2022).

**Proposition 1 (Register neuron induces attention sink and no-op attention)** *Let $u_1 = (W_{in}^{(1)})_{:,1}$ be a register neuron and $u_2 = (W_{out}^{(1)})_{1,:}$ be the corresponding row in the MLP's second weight matrix, with $\|u_2\| \gg \|(W_{out}^{(1)})_{j,:}\|$ for $j \neq 1$. If both $u_1, u_2 \in \text{ker}(W_{in}^{(2)\top}) \cap \text{ker}(W_{head}^{\top})$ and a token $x_s$ is aligned with $u_1$, i.e., $x_s^{(0)} = \beta u_1$ for some $\beta > 0$, then $x_s$ becomes an attention sink and the attention layer has no contribution to the model's output.*

*Proof:*

After the first MLP with a residual connection, the updated hidden states for the tokens are:

$$x_i^{(1)} = x_i^{(0)} + \phi\big(x_i^{(0)} W_{\text{in}}^{(1)}\big) W_{\text{out}}^{(1)}, \qquad i \in \{cls, 1, 2, ..., n\} \tag{2}$$

This can be interpreted as adding a unit vector $v_i \in \mathbb{R}^d$ to the tokens modulated by some coefficient $\alpha_i$:

$$x_i^{(1)} = x_i^{(0)} + \alpha_i v_i, \qquad i \in \{cls, 1, 2, ..., n\} \tag{3}$$

If $x_s^{(0)}$ is aligned with the register neuron $u_1$ (i.e., $x_s^{(0)} = \beta u_1$ for some $\beta > 0$), its activation $\tilde{\alpha}_s = \phi(x_s^{(0)\top} u_1) = \phi(\beta u_1^\top u_1)$ will positively scale the corresponding row in the second MLP matrix $u_2$. As $\|u_2\| \gg \|(W_{\text{out}}^{(1)})_{j,:}\|$ for $j \neq 1$, the update is dominated by the direction of $u_2$, making it the principal contribution to $x_s$ in the residual stream. Thus, $x_s^{(1)} \approx x_s^{(0)} + \tilde{\alpha}_s u_2 = \beta u_1 + \tilde{\alpha}_s u_2$. To make the norm and direction of the update explicit, we rewrite this as $x_s^{(1)} = \beta u_1 + \alpha_s v_2$, where $v_2 = \frac{u_2}{\|u_2\|}$ and $\alpha_s = \tilde{\alpha}_s \|u_2\|$. As the register neuron activation $\|\tilde{\alpha}_s\| \to \infty$, it follows that $\|\alpha_s\|, \|\tilde{\alpha}_s\| \gg \|\alpha_i\|$ for $i \neq s$, and $\|x_s^{(1)}\| \gg \|x_i^{(1)}\|$ for $i \neq s$.

Given identity projection matrices during attention (i.e., the key, queries, values are the tokens themselves), the attention update for the [CLS] token is:

$$x_{cls}^{(2)} = x_{cls}^{(1)} + \sum_{i \in \{cls, 1, ..., n\}} p_{cls,i}\, x_i^{(1)}, \tag{4}$$

where $p_{cls,i}$ is the softmax attention weight assigned by the [CLS] token to token $i$:

$$p_{cls,i} = \frac{\exp\big(x_{cls}^{(1)\top} x_i^{(1)}\big)}{\sum_{j \in \{cls, 1, ..., n\}} \exp\big(x_{cls}^{(1)\top} x_j^{(1)}\big)}. \tag{5}$$

As $\|x_s^{(1)}\| \to \infty$, the dot product $x_{cls}^{(1)\top} x_s^{(1)} = \|x_{cls}^{(1)}\| \cdot \|x_s^{(1)}\| \cdot \cos(\theta)$ grows without bound even for small angle $\theta$ between the two token embeddings, leading the softmax to assign nearly all attention weight to $x_s^{(1)}$. Thus,

$$x_{cls}^{(2)} \approx x_{cls}^{(1)} + x_s^{(1)} = x_{cls}^{(1)} + \beta u_1 + \alpha_s v_2 \tag{6}$$

is the [CLS] token representation after the attention layer.

Since, $u_1, v_2 \in ker(W_{\text{in}}^{(2)\top})$, the [CLS] token representation after the second MLP layer is:

$$x_{cls}^{(3)} = x_{cls}^{(2)} + \phi\Big(x_{cls}^{(2)} W_{\text{in}}^{(2)}\Big) W_{\text{out}}^{(2)} \tag{7}$$

$$= \big(x_{cls}^{(1)} + \beta u_1 + \alpha_s v_2\big) + \phi\big((x_{cls}^{(1)} + \beta u_1 + \alpha_s v_2) W_{\text{in}}^{(2)}\big) W_{\text{out}}^{(2)} \tag{8}$$

$$= x_{cls}^{(1)} + \beta u_1 + \alpha_s v_2 + \phi\big(x_{cls}^{(1)} W_{\text{in}}^{(2)}\big) W_{\text{out}}^{(2)}. \tag{9}$$

Finally, the output prediction is

$$x_{cls}^{(3)} W_{\text{head}} = \big(x_{cls}^{(1)} + \beta u_1 + \alpha_s v_2 + \phi(x_{cls}^{(1)} W_{\text{in}}^{(2)}) W_{\text{out}}^{(2)}\big) W_{\text{head}} \tag{10}$$

$$= \big(x_{cls}^{(1)} + \phi(x_{cls}^{(1)} W_{\text{in}}^{(2)}) W_{\text{out}}^{(2)}\big) W_{\text{head}}, \tag{11}$$

since $u_1, v_2 \in \ker(W_{\text{head}}^{\top})$. This is equivalent to the forward pass without the attention layer, thus exhibiting the "no-op" attention behavior.

**Remarks.** In this proof, we assumed $\|u_2\| \gg \|(W_{\text{out}}^{(1)})_{j,:}\|$ for $j \neq 1$. In practice, we do observe that the register neurons' corresponding rows in the MLP's second weight matrix have specific high norm dimensions (Figure 25).

**Corollary 1 (Register neuron induces implicit attention bias)** *Given the aforementioned set-up, the contribution of register neurons is equivalent to the attention mechanism augmented with explicit bias terms* $(\mathbf{k}', \mathbf{v}')$ *as in Sun et al. (2024):*

$$Attention(Q, K, V; \mathbf{k}', \mathbf{v}') = softmax\big(Q \begin{bmatrix} K^T & \mathbf{k}' \end{bmatrix}\big) \begin{bmatrix} V \\ \mathbf{v}'^T \end{bmatrix}$$

*Proof:*

As our set-up uses identity projection matrices during attention (i.e., the key, queries, values are the tokens themselves), the attention mechanism is:

$$\text{Attention}(Q, K, V; \mathbf{x}', \mathbf{x}') = \text{softmax}\big(X \begin{bmatrix} X^\top & \mathbf{x}' \end{bmatrix}\big) \begin{bmatrix} X \\ \mathbf{x}'^\top \end{bmatrix}. \tag{12}$$

where $X \in \mathbb{R}^{n \times d}$ is the stacked representations of $n$ tokens. From Equation (6), the attention update for the [CLS] token can be decomposed into a contribution from the sink token prior to the first MLP and from the register neuron:

$$x_{cls}^{(2)} \approx x_{cls}^{(1)} + x_s^{(1)} = x_{cls}^{(1)} + \underbrace{\beta u_1}_{\text{pre-MLP sink token}} + \underbrace{\alpha_s v_2}_{\text{register neuron contribution}} \tag{13}$$

Based on the proof for Proposition 1, as the register neuron activation $\|\tilde{\alpha}_s\| \to \infty$, it follows that $\|\alpha_s\|, \|\tilde{\alpha}_s\| \gg \|\alpha_i\|$ for $i \neq s$, and $\|x_s^{(1)}\| \gg \|x_i^{(1)}\|$ for $i \neq s$. In other words, as the register neuron activation grows unboundedly, it dominates the sink token representation and the sink token achieves a significantly larger norm than all other tokens. As $\|x_s^{(1)}\| \to \infty$, the dot product $x_{cls}^{(1)\top} x_s^{(1)} = \|x_{cls}^{(1)}\| \cdot \|x_s^{(1)}\| \cdot \cos(\theta)$ grows without bound even for small angle $\theta$ between the two token embeddings, leading the softmax to assign nearly all attention weight to $x_s^{(1)}$. Thus, the attention update is

$$x_{cls}^{(2)} \approx x_{cls}^{(1)} + x_s^{(1)} = x_{cls}^{(1)} + \beta u_1 + \alpha_s v_2 \tag{14}$$

As the register neuron activation $\|\tilde{\alpha}_s\| \to \infty$, it follows that $\|\alpha_s\|, \|\tilde{\alpha}_s\| \gg \beta$. We take a leading order approximation to simplify Equation (14) to

$$x_{cls}^{(2)} \approx x_{cls}^{(1)} + \alpha_s v_2 \tag{15}$$

A similar update rule holds for any arbitrary token $i$:

$$x_i^{(2)} = x_i^{(1)} + \alpha_s v_2, \qquad i \in \{cls, 1, 2, ..., n\} \tag{16}$$

This update can be achieved by plugging in the register neuron contribution $\alpha_s v_2$ into $\mathbf{x}'$ in Equation (12):

$$\text{Attention}(Q, K, V; \alpha_s \mathbf{v_2}, \alpha_s \mathbf{v_2}) = \text{softmax}\left(X \begin{bmatrix} X^\top & \alpha_s \mathbf{v_2} \end{bmatrix}\right) \begin{bmatrix} X \\ \alpha_s \mathbf{v_2}^\top \end{bmatrix}. \tag{17}$$

Using a similar argument from the proof for Proposition 1, as $\|\alpha_s\| \to \infty$, for any token index $i$, the dot product $x_i^{(1)\top}(\alpha_s v_2) = \|x_i^{(1)}\| \cdot \|\alpha_s v_2\| \cdot \cos(\theta)$ grows without bound even for small angle $\theta$ between the two vectors, leading the softmax to assign nearly all attention weight to $\alpha_s v_2$. Thus, the update representation for token $i$ after this augmented attention is:

$$x_i^{(2)} = x_i^{(1)} + \alpha_s v_2, \qquad i \in \{cls, 1, 2, ..., n\} \tag{18}$$

which is equivalent to Equation (16), thus showing that register neurons induce a bias term in the attention mechanism. This suggests that we can directly plug in the contribution of register neurons into the attention mechanism as in eq. (17), without the need for a test-time register. We investigate this in the next section.

### A.2 Removing outliers with attention biases

While we focus on shifting outliers outside the image in this work, high-norm outliers still persist in the extra test-time register. Here, we propose a preliminary method to eliminate these artifacts without additional register tokens, by using test-time attention biases to fully remove high-norm outliers from the residual stream.

**Incorporating attention biases at test time.** Sun et al. (2024) observed that language and vision Transformers have massive values at certain dimensions and proposed to remove them by incorporating attention biases during training. Specifically, given the query, key, and value matrices $Q, K, V \in \mathbb{R}^{T \times d}$ for each attention head, they train additional parameters $\mathbf{k}', \mathbf{v}' \in \mathbb{R}^d$ such that

$$\text{Attention}(Q, K, V; \mathbf{k}', \mathbf{v}') = \text{softmax}\left(\frac{Q \begin{bmatrix} K^T & \mathbf{k}' \end{bmatrix}}{\sqrt{d}}\right) \begin{bmatrix} V \\ \mathbf{v}'^T \end{bmatrix}$$

We propose to create these attention biases in OpenCLIP ViT-B/16 training-free by approximating the effect of a test-time register on the attention computations. To set $\mathbf{k}'$ and $\mathbf{v}'$ for each attention head, we compute the mean key and value vectors of a test-time register token across 1000 images. At test-time, we zero out the activations of register neurons using the parameters from Section 3.2.

**Results.** We report zero-shot ImageNet performance in Table 7. Zeroing out the register neurons leads to a significant performance drop, but this drop is fully recovered by introducing the attention bias, which restores the model's original performance. This suggests that zeroing out registers effectively removes outliers, while the attention bias approximates their influence at test time. Importantly, we no longer observe high-norm outliers across all layers (Figure 8). We also find that attention biases create clean attention maps free of artifacts comparable to using a test-time register (Figure 9). Our findings support the claim by Sun et al. (2024) that registers primarily function as attention biases. Although massive activations are relatively modest in Vision Transformers (e.g., $< 500$), they can exceed the mean activation by up to $10,000\times$ in language models, creating challenges for quantization—where specialized methods are often needed to handle such outliers (Xiao et al. (2023)). Our results suggest that these large activations can be controlled without any additional training overhead. We leave the application of our method to the language domain for future work.

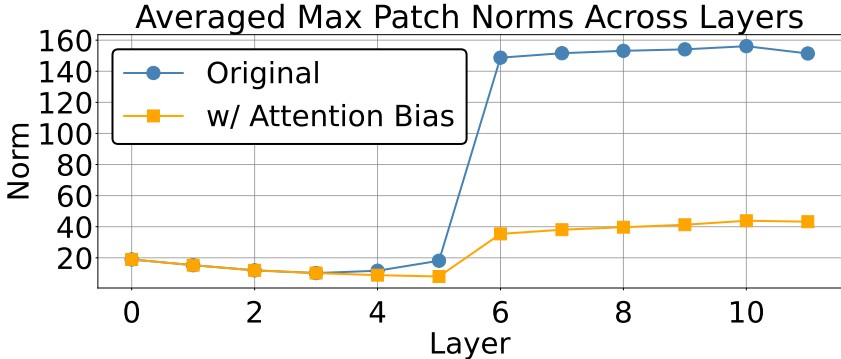

Figure 8: **Averaged max patch norm across all layers in OpenCLIP-B/16 using attention biases.** We find that using attention biases completely removes the presence of high-norm outliers from the residual stream.

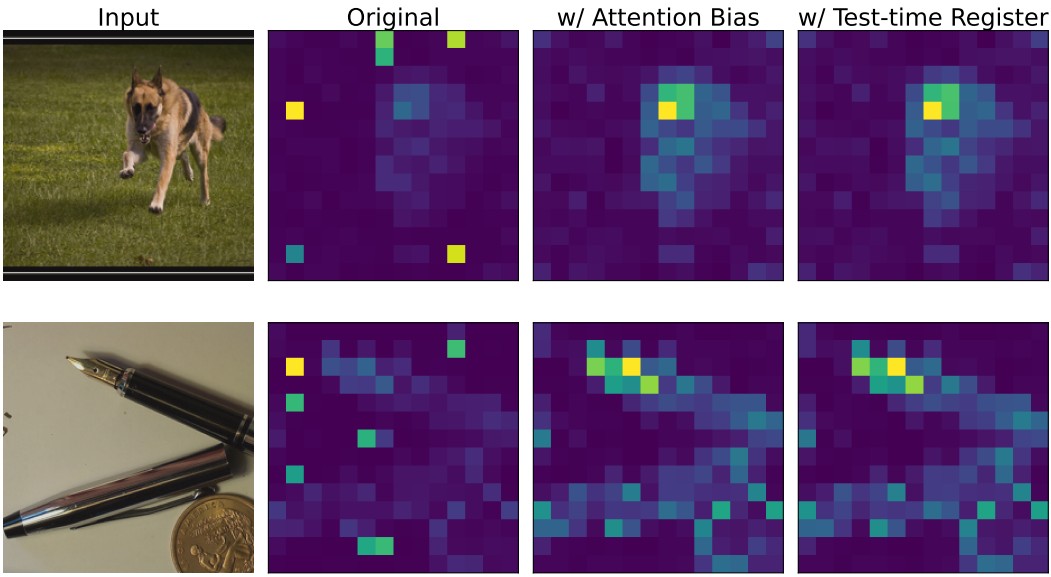

Figure 9: **Using attention biases produces similarly clean attention maps as using a test-time register.** We compute the attention map of the `[CLS]` token averaged across the attention heads of the last layer on OpenCLIP-B/16, finding that both attention biases and test-time registers remove artifacts.

| OpenCLIP | Acc. |
| --- | --- |
| ViT-B/16 | 71.3 |
| w/ zeroed register neurons | 55.6 |
| w/ attention bias | 71.3 |

Table 7: **Creating attention biases at test-time maintains zero-shot ImageNet performance.** Zeroing out register neurons significantly drops performance, but attention biases recover the loss.

### A.3 Algorithm overview

We provide a broad overview of Algorithm 1 and Algorithm 2 accompanied by illustrations of their execution. To find register neurons with Algorithm 1, we compute the average activation of each neuron on high-norm patches across an image set and return the $top\_k$ neurons with the highest values (Figure 10). Using these discovered register neurons, we implement a test-time register by initializing a dummy token to a vector of zeros which is passed forward along with the regular tokens. According to Algorithm 2, whenever a register neuron is encountered during the forward pass, the maximum register neuron activation across the tokens is copied into the test-time register and the activations of this register neuron are zeroed-out elsewhere (Figure 11).

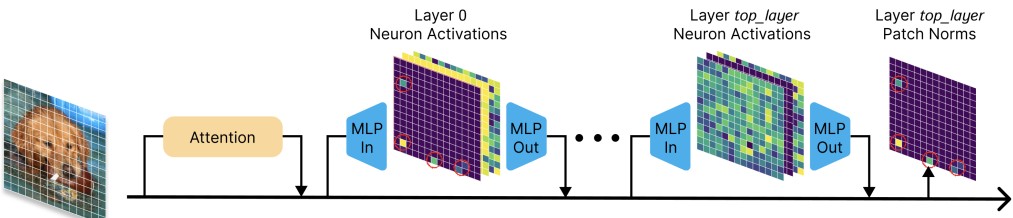

Figure 10: **Finding register neurons.** Our register neuron discovery algorithm returns the neurons with the highest activations on outlier patch locations.

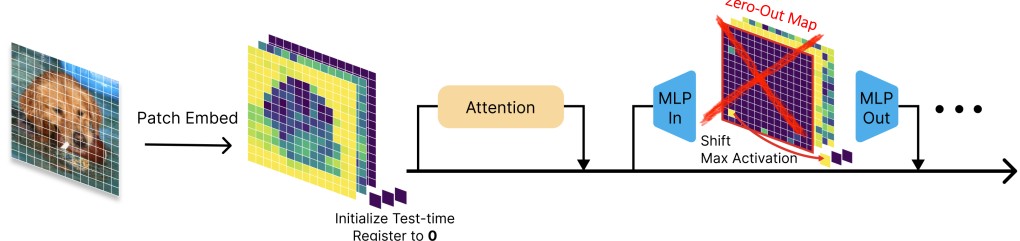

Figure 11: **Test-time register intervention.** During the forward pass, the max activation from each layer's register neuron is shifted to the test-time register. The register neuron activations are then zeroed-out over the other tokens, and the inference computation proceeds.

---

**Algorithm 1** FINDREGISTERNEURONS

1: **Input:** Image set $\mathcal{I} = \{I_1, \ldots, I_M\}$, maximum layer index $top\_layer$, number of register neurons to return $top\_k$, number of neurons in each layer $N$
2: **Output:** $top\_k$ register neurons
3: $avg\_act \leftarrow \mathbf{0}_{top\_layer \times N}$ # initialize array for average activations
4: **for all** $I_i \in \mathcal{I}$ **do**
5: $\quad O \leftarrow \text{FindOutliers}(I_i)$ # get indices of top-norm patches
6: $\quad$ **for** $\ell = 0$ to $top\_layer$ **do**
7: $\quad\quad$ **for** $n = 0$ to $N - 1$ **do**
8: $\quad\quad\quad avg\_act[\ell, n] \leftarrow avg\_act[\ell, n] + \frac{1}{|O|M}\sum_{p \in O} \text{activation}_{\ell,n}(I_i, p)$ # avg. activation of neuron on outlier patches
9: $\quad\quad$ **end for**
10: $\quad$ **end for**
11: **end for**
12: **return** $top\_k$ neurons with largest $avg\_act$

**Algorithm 2** SHIFTREGISTERNEURONS

1: **Input:** List of register neuron indices in this layer $\mathcal{R}$, array of $n$ neuron activation maps over $T$ tokens produced by this layer $A \in \mathbb{R}^{T \times N}$
2: **Output:** updated $A$
3: **for all** $r \in \mathcal{R}$ **do**
4: $\quad A[-1, r] \leftarrow \max_{t \in [0, T-1]} A[t, r]$ # set test-time register token (idx = -1) to max register neuron activation value over all tokens
5: $\quad A[0:T-1, r] \leftarrow 0$ # zero-out register neuron activation for all other tokens
6: **end for**
7: **return** $A$ # proceed with forward pass

## A.4 Hyperparameter Selection

Algorithm 1 relies on three hyperparameters: (1) the outlier threshold, (2) the index of the highest layer we search up to when identifying register neurons, and (3) the number of register neurons to return. The first two are straightforward to set using simple heuristics. Specifically, the outlier threshold can be defined using the classical criterion of three standard deviations above the mean patch norm. The second, $top\_layer$, is chosen as the layer at which outliers first emerge. Setting the number of neurons to return is slightly more involved, relying on sweeping $top\_k$ until the outliers are suppressed. However, this is manageable in practice since the number of register neurons is sparse. We test how scalable this approach is in Section A.7.

## A.5 Ablating number of register tokens

In the main text, we focused on evaluation using a single register. Here, we examine the impact of employing multiple registers. Following the analysis from Sun et al. (2024), we separate the influence of high-norm tokens on the attention output (i.e. after multiplication with value features). The attention output at each token $t$ can be decomposed into two components: value contributions from the register tokens $\mathcal{R}$, and value contributions accumulated over the [CLS] and patch tokens.

$$\text{Attention}(Q, K, V)_t = \sum_i p_i^t v_i = \underbrace{\sum_{i \in \mathcal{R}} p_i^t v_i}_{\text{registers contribution}} + \underbrace{\sum_{i \notin \mathcal{R}} p_i^t v_i}_{\text{non-registers contribution}} \tag{19}$$

where $p_i^t$ is the attention weight of query token $t$ to token $i$, and $v_i$ is the value embedding associated with token $i$.

**Experimental setup for test-time registers.** Given Equation (19), we analyze the contribution of test-time registers to the value updates of all other tokens in the final attention layer of DINOv2 ViT-L/14. We first compute the value update for each token across the ImageNet validation set using a single test-time register. Then, we vary the number of registers and, for each setting, compute the cosine similarity between the resulting value update and the one obtained with a single register, averaged over all tokens. Since all test-time registers are initialized to zero, we break symmetry by randomly assigning which test-time register receives the outlier register neuron activation. Thus, each test-time register holds the activations from a different set of register neurons.

**Increasing the number of test-time registers beyond one has a negligible impact.** Our results in Table 8 show that increasing the number of registers has a marginal impact on the internal computation of the model. We see the qualitative effect of this on the last layer's [CLS] token attention map averaged across all heads in Figure 12. One test-time register removes all the high-norm artifacts from the original model's attention map. We then evaluate the impact on downstream performance using a linear probe on the NYUd depth estimation task. Overall, we find that adding more test-time registers beyond one does not significantly change performance. We note that there is a slight degradation in performance beyond three test-time registers, although performance remains higher than without registers. These results align with the finding from Sun et al. (2024) that register tokens act as implicit attention biases. Thus, a single register token can be sufficient to induce the

| # Test-time Registers | Cosine Sim. w.r.t. 1 Register |
|---|---|
| 1 | 1.000 |
| 2 | 0.998 |
| 3 | 0.995 |
| 4 | 0.991 |
| 5 | 0.985 |

Table 8: **Cosine similarity between value updates from one test-time register and increasingly larger sets of register tokens.** Increasing register count beyond one has marginal effect.

| # Test-time Registers | NYUd rmse $\downarrow$ |
|---|---|
| 0 | $0.3876 \pm 0.0014$ |
| 1 | $0.3774 \pm 0.0013$ |
| 2 | $0.3771 \pm 0.0014$ |
| 3 | $0.3785 \pm 0.0014$ |
| 4 | $0.3791 \pm 0.0017$ |
| 5 | $0.3795 \pm 0.0016$ |

Table 9: **NYUd RMSE for DINOv2-L/14 with varying numbers of test-time registers.** Increasing the number of registers beyond one has minimal gains.

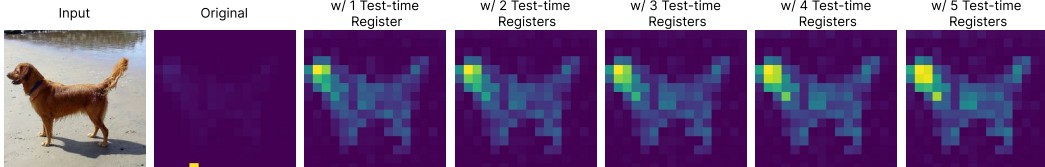

Figure 12: **One test-time register is sufficient to remove artifacts.** We increment the number of test-time registers where each register holds the activations from a different set of register neurons. The last layer's average `[CLS]` token attention map from DINOv2 ViT-L/14 does not significantly change with more test-time registers.

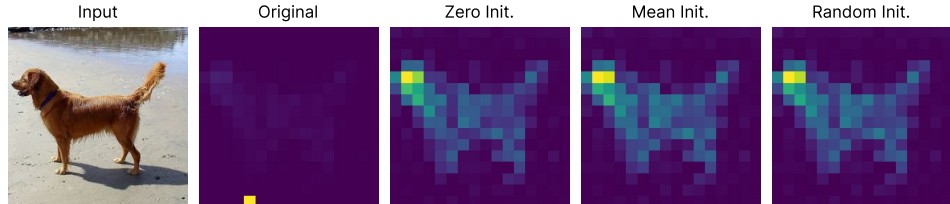

Figure 13: **Different test-time register initialization strategies yield similar attention maps.** We experiment with three different initialization strategies and find that they do not impact the test-time register's ability to hold high norms and clean up attention maps.

same value update across tokens and clean up internal features. We further corroborate this behavior on trained registers next.

**Experimental setup for trained registers.** We use DINOv2 ViT-L/14 trained with four registers and calculate the value update contributions from each of the four registers to both image patch tokens and the `[CLS]` token. To approximate their collective effect with a single token, we construct a synthetic register by assigning, for each embedding dimension, the value (including sign) with the largest absolute magnitude across the four trained registers. We then compute the value update induced by this synthetic register and measure its cosine similarity with the original update from the four-register setup, averaged over all tokens.

**Multiple trained registers can be approximated by one token.** We obtain a cosine similarity of $0.834$ between the value update from four registers and from the one constructed token. This suggests that a single register can retain the majority of the representational effect of multiple trained registers. This suggests that the main role of trained registers may be holding large activations. However, since the cosine similarity is not $1.0$, this implies that the registers may also involve additional mechanisms, beyond simply holding large activations, in contributing to the model's behavior.

### A.6 Evaluating different initial values for test-time registers

**Test-time register initialization strategies.** We experiment with various initialization strategies for the test-time registers and evaluate their performance through linear probing on ImageNet, CIFAR10, and CIFAR100 classification tasks. Specifically, we test three initialization methods: initializing the token to zero, initializing it with a Gaussian distribution matching the mean and standard deviation of the patch tokens, and initializing it to the mean of the patch tokens.

**Different initializations produce similar results.** All three approaches yield similar probing performance as reported in Table 10. We attribute this outcome to the fact that the primary contribution of the register during attention comes from the large activations it holds, while the other values, which are much smaller in magnitude, have little impact on the final result. In Figure 13, we visualize the last layer's average `[CLS]` token attention map from DINOv2 ViT-L/14, and observe that different initialization strategies do not significantly impact the test-time register's ability to remove artifacts.

|  | IN1k | CF10 | CF100 |
|---|---|---|---|
| `[reg]` (zero init.) | 84.5 | 99.1 | 93.0 |
| `[reg]` (rand. init.) | 84.6 | 99.2 | 92.8 |
| `[reg]` (mean init.) | 84.5 | 99.1 | 92.9 |

Table 10: **Image classification via linear probing the test-time register token (DINOv2 ViT-L/14).** We sweep over different initialization strategies for the token and find that they yield similar results.

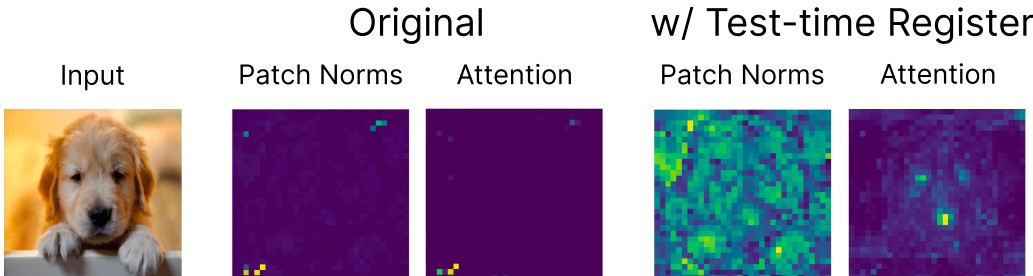

Figure 14: **Mitigating artifacts in InternViT-6B. Using a test-time register removes artifacts from the** `[CLS]` **attention map.**

|  | VOC 2007 | VOC 2012 | COCO 20k |
|---|---|---|---|
| InternViT-6B | 30.3 | 35.4 | 22.4 |
| w/ test-time register | 41.6 | 48.7 | 32.7 |

Table 11: **Unsupervised Object Discovery with LOST** (Siméoni et al., 2021). Using a test-time register boosts LOST performance on InternViT-6B, one of the largest open-source ViTs.

### A.7 InternViT-6B

To test the scalability of our approach, we apply our analysis from Section 3 to InternViT-6B, one of the largest publicly available vision transformers (Zhu et al., 2025). We find that outliers form at layer 29 (out of 45). We then tracked the max norms across image patches at the output layer, and found that outlier patch norms reach up to a value of 7000, while the median patch norm value is 332 and the standard deviation is 396. Using Algorithm 1, we then identified 300 register neurons out of a total of 576,000 candidate neurons responsible for driving outlier formation. When we apply a test-time register with these register neurons, the maximum output patch norm at the output layer only reaches 608 on average, effectively mitigating the outliers.

To assess the downstream impact, we ran unsupervised object discovery using the LOST algorithm (as in Section 5.3) since Darcet et al. (2024) found that the performance on this task correlates with the smoothness of a model's attention maps. We report the correct localization (corloc) results below with and without a test-time register on InternViT-6B, and observe up to a 13-point correct localization improvement.

### A.8 Typographic attacks

Darcet et al. (2024) showed that outlier patches retain less local information by demonstrating that their pixel/position information is harder to recover with a linear probe compared to normal patches. However, this analysis leaves open the possibility that local information at outliers might still be redistributed to other patches and indirectly influence the `[CLS]` token. We demonstrate here that this is not the case and that outlier tokens mask local patch information. To evaluate this masking effect, we use *typographic attacks* as a test-bed. Azuma & Matsui (2023) showed that CLIP is vulnerable to typographic attacks, where images can be misclassified based on written text in the image rather than the depicted object (Figure 15). We strategically place high-norm artifacts within an image to mask out adversarial patches while preserving semantic content. This intervention is highly localized and relies on activating only a small fraction of neurons to produce targeted changes in the image representation.

**Experimental setup.** Following Azuma & Matsui (2023), we use OpenAI CLIP (Radford et al., 2021) and identify register neurons with Algorithm 1 on this model. Then, we localize the largest region of text in the image with OCR and, using the register neurons, move high-norm outliers onto the patches corresponding to the text location. We evaluate on the RTA-100 dataset (Azuma & Matsui, 2023), which contains approximately 1000 images with text written on top of an object from one of 100 classes. We calculate zero-shot accuracy by comparing CLIP's image embedding to the "real" and "attack" labels' text embedding.

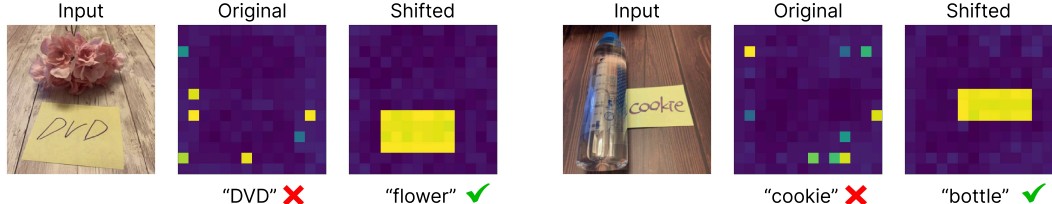

Figure 15: **Qualitative results on typographic attacks.** We show the patch norms of the last layer before ("Original") and after ("Shifted") intervening on register neurons. Shifting the outliers to the text location masks the text in activation space and results in more accurate classification.

**Results.** Qualitative results in Figure 15 show that shifting the activation maps of ten register neurons early in the model's computation causes outlier patches to later appear corresponding to the text area. Table 12 presents the attack success rate and find that it significantly drops after our intervention, indicating that our method masks part of the model's internal representation without harming semantic content. Our results suggest that patch information at outliers is lost from the model computation rather than transferred elsewhere. We also present results

| Model | Attack success % |
|---|---|
| CLIP | 50.5 |
| w/ pixel ablation | 7.6 |
| w/ register neuron edit | 7.5 |
| w/ random neuron edit | $50.5 \pm 0.2$ |

Table 12: **Typographic attack success rate.** We leverage register neurons to shift outliers to tokens at areas with text, effectively masking the adversarial text out in activation space.

of an alternative masking procedure in the *input* space, setting areas corresponding to text to the mean pixel value of the region. Our intervention matches this method's performance using only a sparse modification – repurposing CLIP's internal mechanisms by modifying roughly $0.02\%$ of all neurons, compared to masking $\sim 10\%$ of the input. This highlights the influence of register neurons in guiding CLIP's visual representation and output behavior – shifting outliers on top of the text areas is performance-wise equivalent to the text never being present from the start. In contract, editing three random sets of ten neurons has minimal effect (w/random neuron edit). The finding that the register neurons effectively mask patch information explains why mitigating the artifacts can improve dense prediction tasks.

### A.9    Attention maps after intervening upon register neurons

We present attention maps from all layers for several example images, applying our intervention on OpenCLIP (Figure 16) and DINOv2 (Figure 17 and Figure 18). Our results show that intervening upon register neurons produces clean attention maps free of the high-norm artifacts.

### A.10    Additional activation maps of register neurons

While our primary focus is on layer 6 of OpenCLIP (Section 3.1), we also include additional activation maps highlighting neurons that strongly activate at the top outlier location. As shown in Figure 19, the most active neurons from earlier layers produce activation maps that align closely with the output patch norms. This consistent alignment supports our hypothesis in Section 3.2 that a small number of sparsely activating neurons determine outlier locations even before they fully emerge. In OpenCLIP, we observe that top-activating neurons in earlier layers causally influence those in later layers. In particular, interventions on neurons in layer 5 or earlier lead to automatic changes in layer 6 activations (see Figure 4).

### A.11    Additional OpenCLIP experiments

#### A.11.1    How do outliers emerge in ViTs?

**A small subset of neurons have a consistently high contribution at the top outlier position.** To localize the set of neurons that drive outlier formation, we evaluate the norm contribution of individual neurons at layer 6, where outliers appear. In Figure 20, we plot the distribution of activations for all layer 6 neurons, comparing the average activation at a top outlier patch with a randomly selected non-outlier patch. We observe that five MLP neurons consistently have large activations on the top

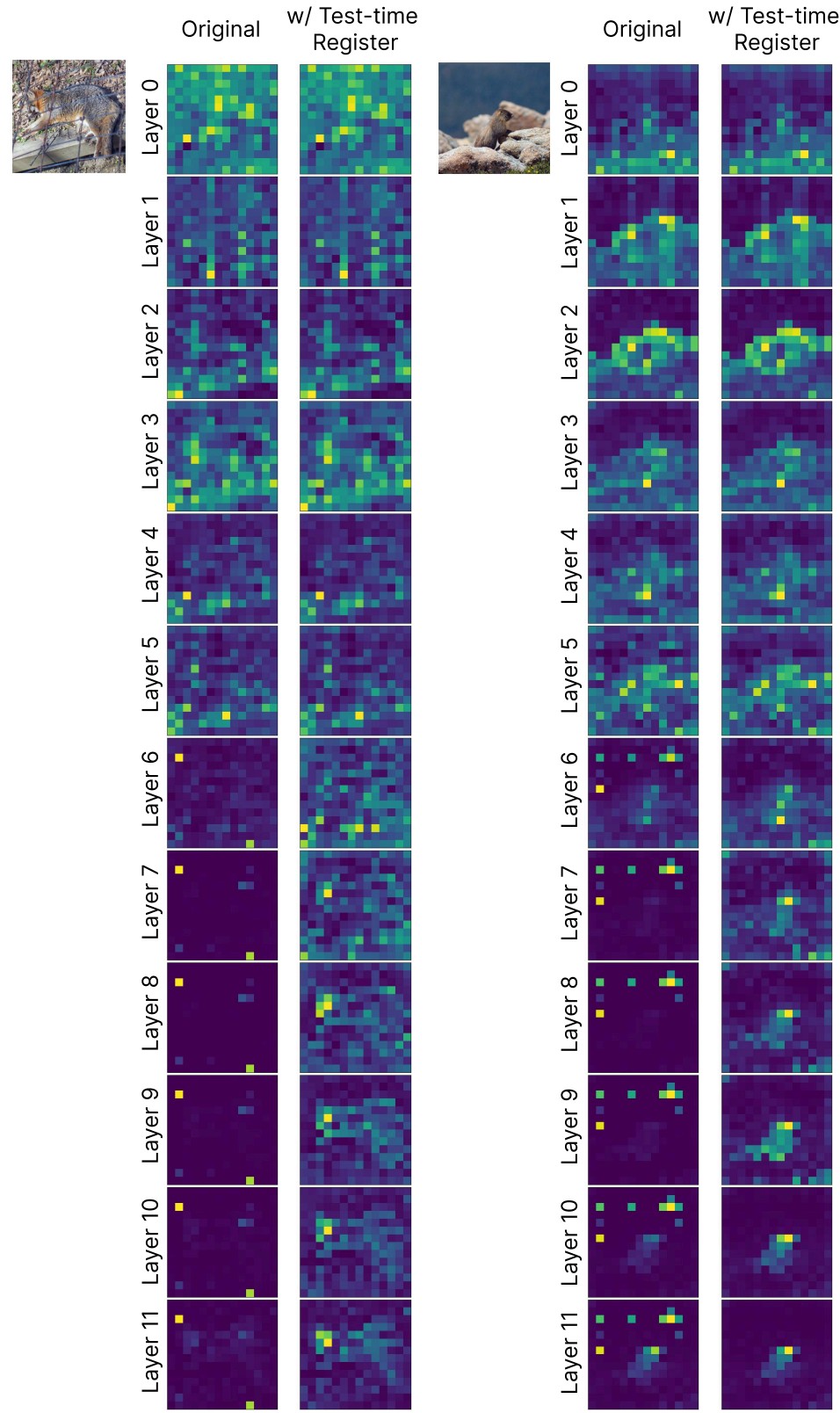

Figure 16: **Attention maps for OpenCLIP with test-time register.** We show the mean [CLS] attention maps from all layers for several input images. Outliers appear in layer 6 for the original model but do not appear with test-time registers, producing clean, interpretable attention maps.

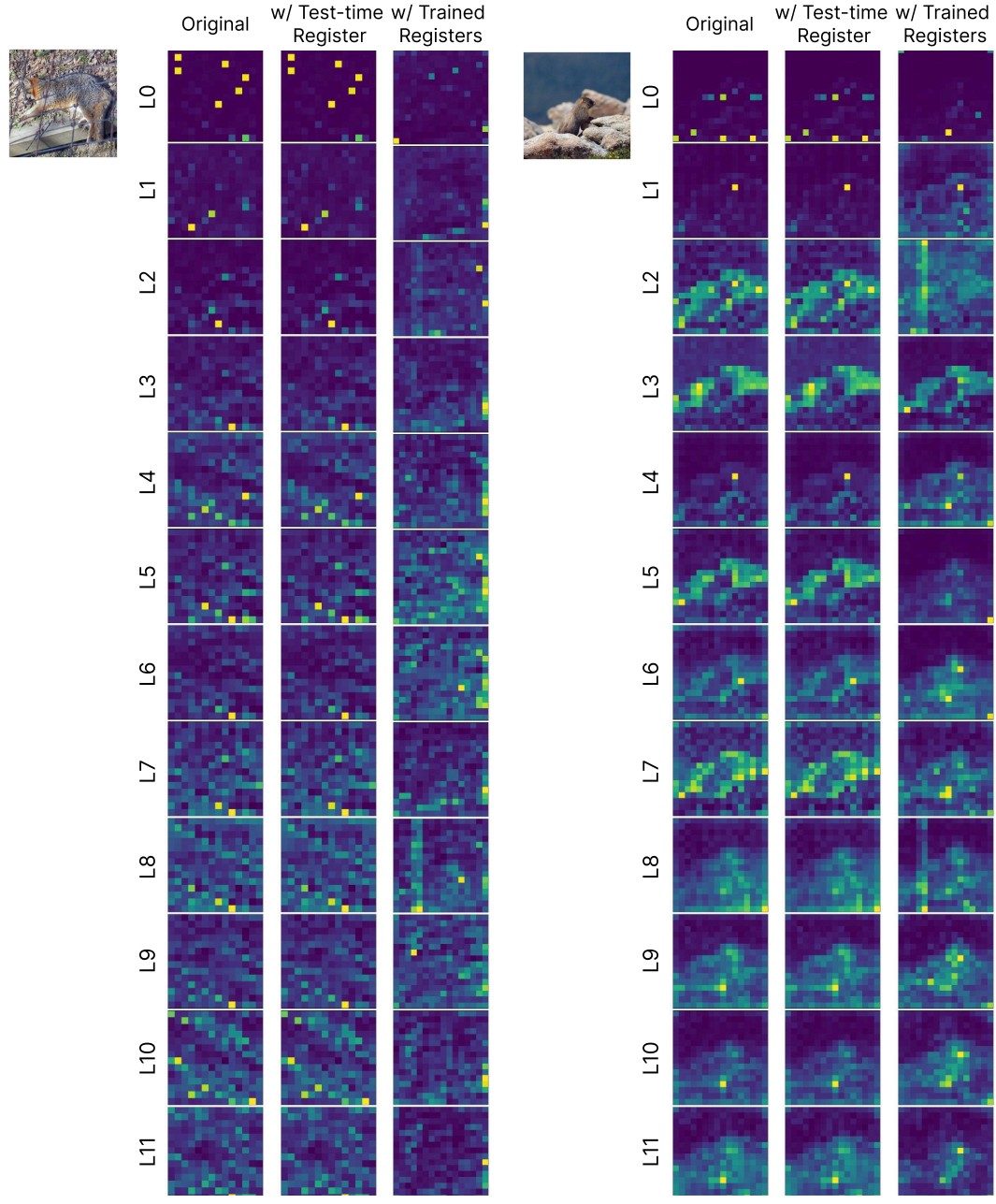

Figure 17: **Attention maps for DINOv2 with test-time register (Layers 0-11).** We show the mean `[CLS]` attention maps from layers 0-11 for several input images. As we do not intervene in these layers, there is no difference in the attention maps.

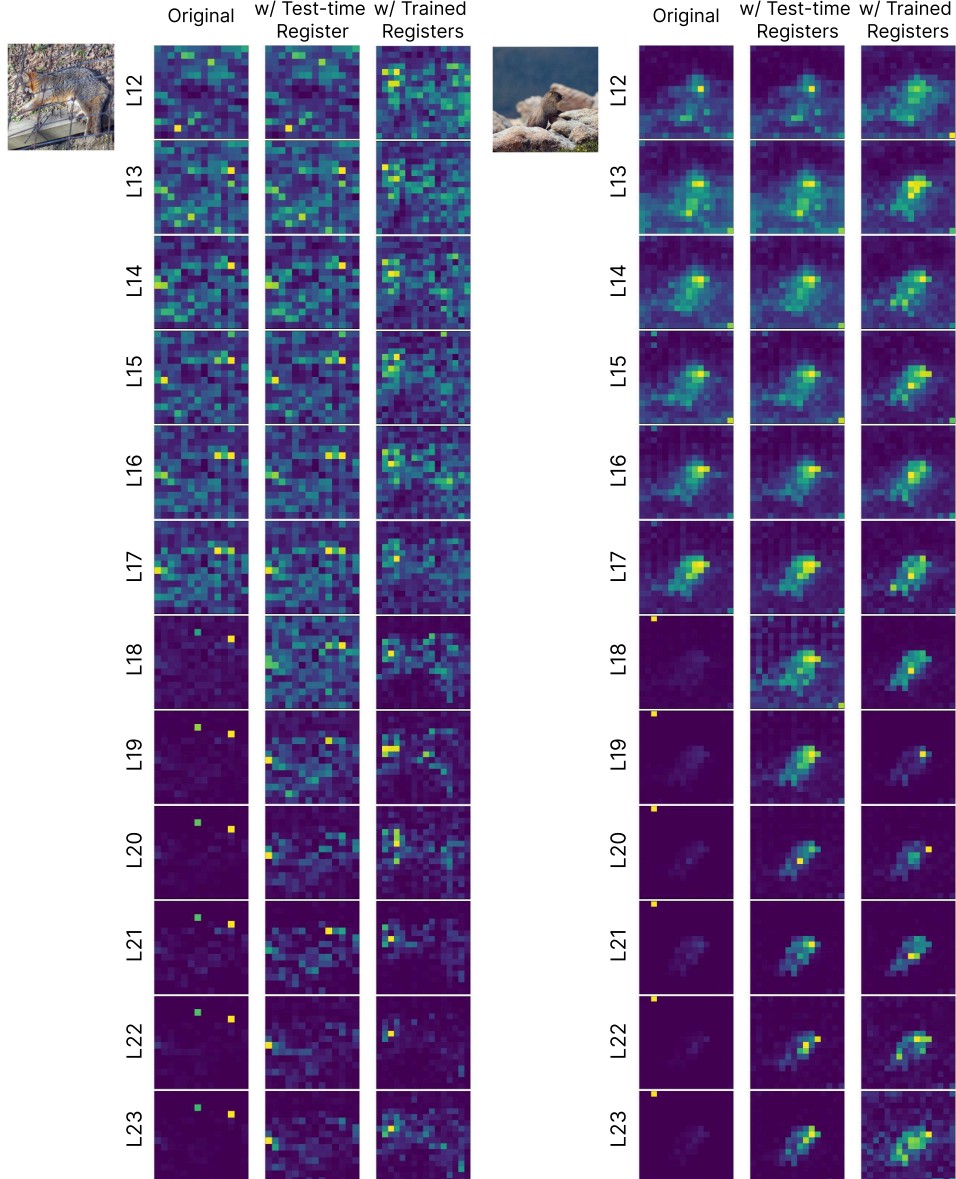

Figure 18: **Attention maps for DINOv2 with test-time register (Layers 12-23).** We show the mean `[CLS]` attention maps from layers 12-23 for several input images. Artifacts appear in uniform regions around layer 18 for the original model. With test-time registers, attention maps become clean and reveal the images' main objects.



Figure 19: **Additional activation maps of neurons that activate highly on the top outlier position.** Even in layers before outliers emerge (layer 6), we observe that there exist neurons whose activation maps closely align with the output patch norms.

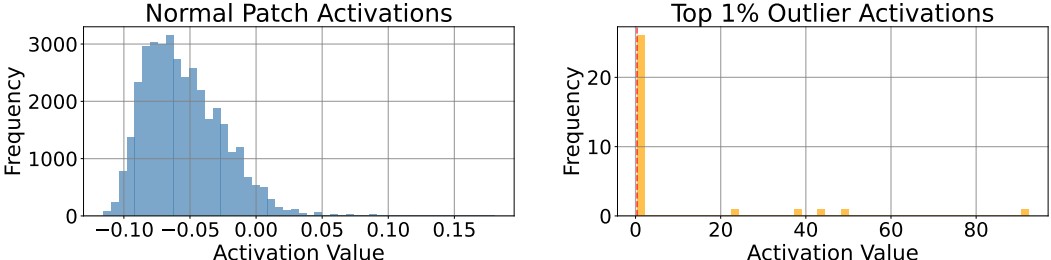

Figure 20: **Neuron activation distributions differ between outlier and non-outlier patches**. For all layer 6 neurons, we average their activations on the top outlier patch and a randomly selected non-outlier patch across 1000 images. Non-outlier patches have a more symmetric distribution (left), whereas outlier patches show a skewed distribution with most activations near zero. Five neurons consistently exhibit high activations for outlier patches across images (right).

outlier, creating a skewed distribution, and low activations on non-outliers. To measure the effect of these neurons, we track their aggregate contribution to the residual stream over 1000 images by computing the norm and pairwise cosine similarities of their update vectors. As shown in Figure 21, these neurons have consistent, high-norm contributions with effectively constant directions. In contrast, a random set of five neurons produces low-norm updates whose directions vary substantially.

### A.11.2 Adding register at test-time

In Section 4, we moved outlier activations to an added, test-time register in OpenCLIP and observed that it removes outliers from the image patches. After moving the outliers to a single test-time register, we measured the test-time register's output norm, the maximum norm of image patch outputs, and the highest last-layer `[CLS]` attention, both to the test-time register and any image patch. We now present the full results in Figure 22 and find a nearly identical distribution to Figure 4, indicating that the image patch outliers have been moved to the test-time register.

### A.11.3 Intervening on random neurons

**Intervening on random neurons is ineffective for shifting outliers.** While register neurons are highly effective for shifting outliers (Section 3), we evaluate a random baseline to further justify our selection of neurons. We perform a similar intervention method used for register neurons on random neurons in OpenCLIP ViT-B/16, but we find they are ineffective for shifting outliers. For each layer, we randomly select the same number of neurons as those in our corresponding set of register neurons. During the forward pass, we intervene on the activation maps of these random neurons. We copy the highest activation across the original activation map to a randomly selected patch. In Figure 23, we observe that the selected patch does not absorb the high norms, demonstrating that using random neurons fails to shift outliers.

**Intervening on random neurons with high activations is similarly ineffective**. We also test the hypothesis that it is the *high activations* specifically of register neurons that cause the outliers to appear. Instead of using the highest activation for the random neuron, we instead copy the highest activation from the corresponding register neuron. However, we find similarly ineffective results in

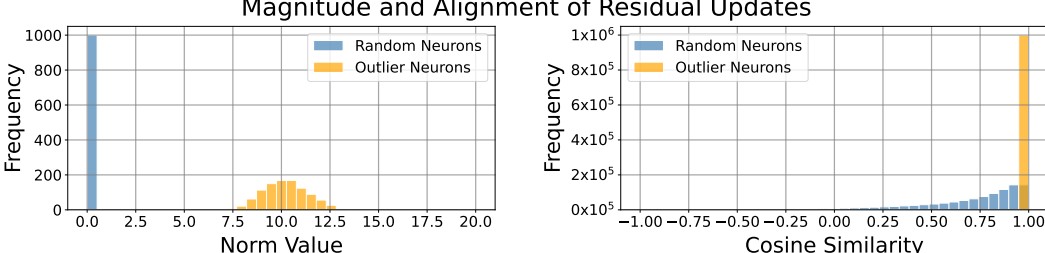

Figure 21: **Outlier neurons show stable, high-magnitude contributions.** We track the overall residual stream update from five random neurons and five identified outlier neurons on the top outlier patch across 1000 images. Outlier neurons contribute high norm updates (left) that are consistent in direction (right).

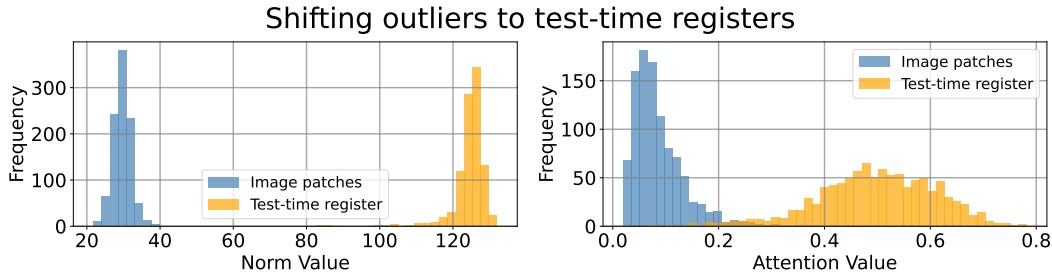

Figure 22: **Intervening on activations of register neurons effectively shifts outliers to test-time registers.** For all register neurons, we copy their highest activation into the test-time register and zero out the activations elsewhere. The test-time register absorbs the high norms and attention from the `[CLS]` token, indicating that the image patches no longer have outliers.

Figure 24. The inability of random neurons to shift outliers reinforces our hypothesis that register neurons (i.e., their decoder direction) specifically are crucial for outlier emergence.

**Decoders of register neurons have large weights in certain dimensions.** Sun et al. (2024) observe that outlier patches have massive activations in certain dimensions. Given this observation, we extract the decoder weights (post-non-linearity) for four register neurons from layer 5 and plot the distribution of weight magnitudes across dimensions in Figure 25. We find that certain dimensions (e.g. 579, 408) consistently have large weights across register neurons, which we do not observe with other random neurons. This property further supports the hypothesis that register neurons are the key mechanism that leads to outlier formation.

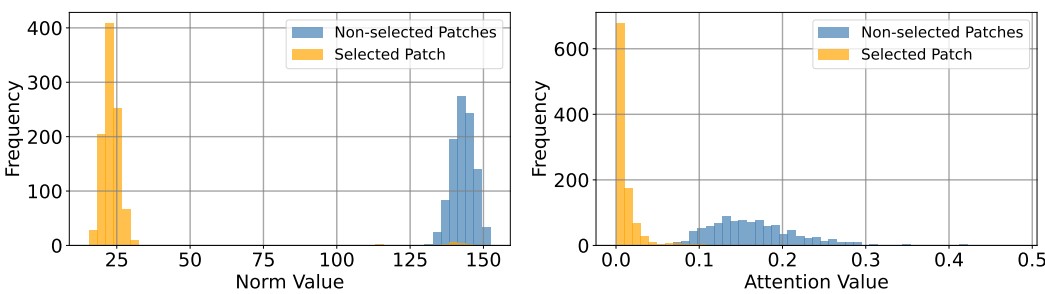

Figure 23: **Intervening on random neurons does not shift outliers.** We attempt to apply our intervention method with random neurons to move outliers to arbitrary patches. However, we find that the selected patch fails to absorb the high norms, suggesting that our selection of register neurons is meaningful.

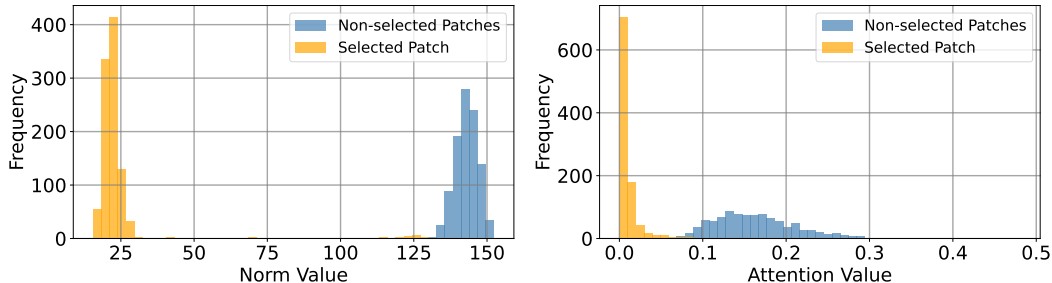

Figure 24: **Copying in high activations into random neurons does not shift outliers.** To verify that it is the high activations in the corresponding register neuron channels that create outliers, we intervene upon random neurons but copy in the highest activation value from a corresponding register neuron. We find that the norm of the selected patch remains low, supporting our selection of register neurons to intervene upon.

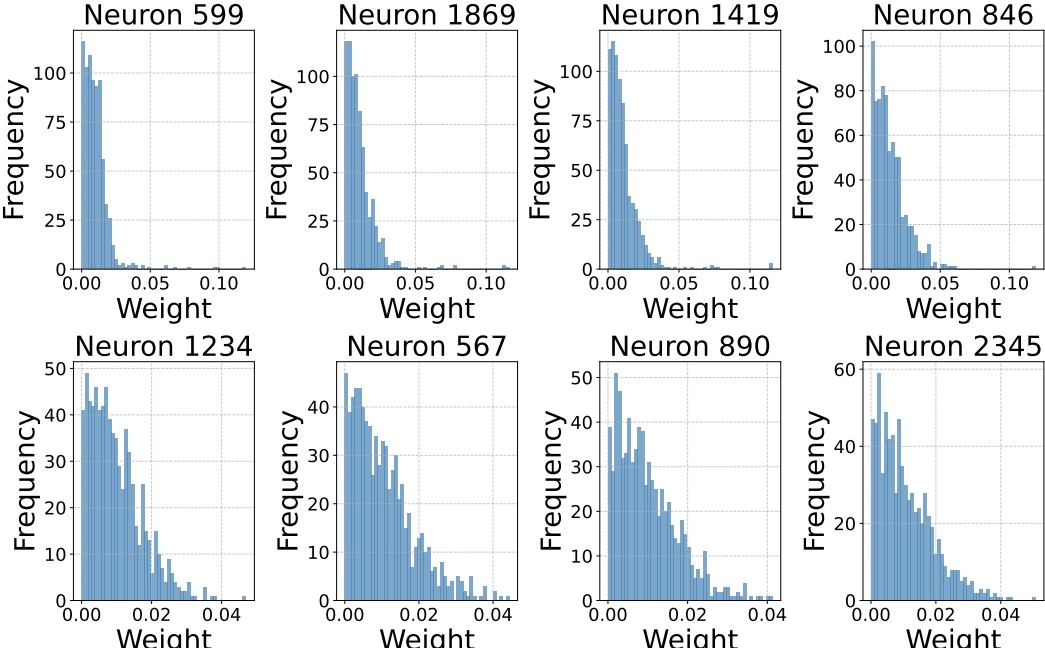

Figure 25: **Decoder weights of register neurons have large values in specific dimensions.** We take the absolute value of decoder weights from layer 5 neurons in OpenCLIP and plot the distribution of magnitudes across dimensions. Top row: layer 5 register neurons. Bottom row: layer 5 random neurons.

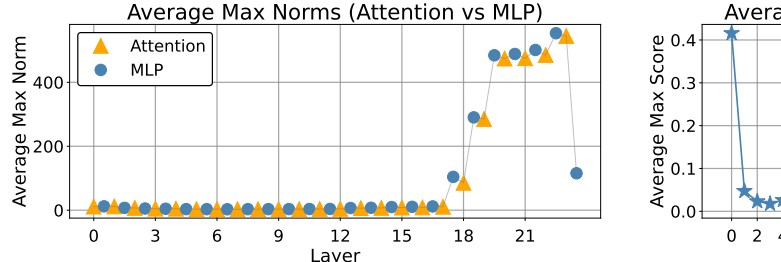 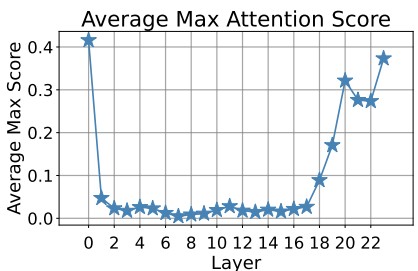

Figure 26: **Outlier patches appear after MLPs in DINOv2; attention sinks appear after outlier patches.** Left: Max norms across image patches (DINOv2-L/14). Right: max attention scores of the [CLS] token in the last layer. In both plots, we average across 1000 images. The increase in max norms and emergence of attention sinks occur in consecutive layers.

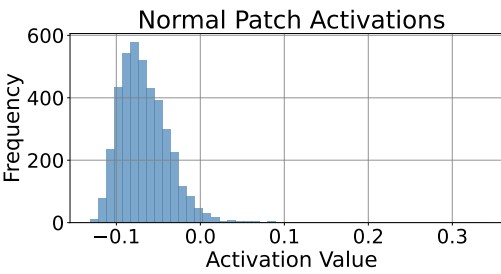 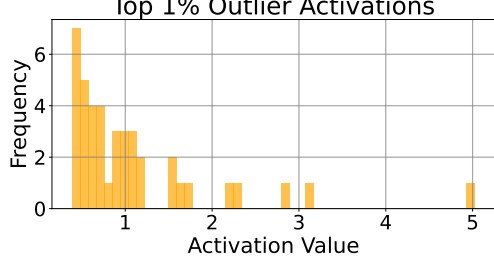

Figure 27: **A small set of neurons in DINOv2 have high activations on the top outlier patch (right).** We average the neuron activations at layer 17 in DINOv2 for the top outlier patch and a randomly selected, non-outlier patch. Both types of patches have skewed distributions. A small subset of neurons (<25) consistently exhibit high activations for outlier patches across images (right).

## A.12  Investigating outliers in DINOv2

We outline a similar investigation as presented in Section 3 on DINOv2-L/14 and find a sparse set of neurons that can be used to move outliers to arbitrary patches.

**Outlier patches appear after MLPs.** To evaluate whether the attention block or MLP causes outliers to form, we plot the maximum norm of the residual stream across all patches for every attention and MLP component. In Figure 26, we observe that the outliers appear at the MLP of layer 17, with attention sinks emerging soon after. These observations mirror Figure 2, where we also find a single layer in OpenCLIP that outliers tend to start appearing.

**A small subset of neurons shows consistently high activations before outlier patches.** To investigate layer 17, we evaluate the distribution of activations for its MLP neurons on the top outlier patch, averaged across 1000 images. To identify the top outlier patch, we find the patch with the maximum norm in the 2nd-to-last layer's output (layer 22). We choose the 2nd-to-last layer to identify outliers because the maximum patch norm drops in the last layer output (Figure 26), making it difficult to differentiate outliers from normal patches. In Figure 27, we observe that the activation distribution for outliers is heavily skewed, with <25 neurons having disproportionately high activations. We find a similar skew in OpenCLIP (Figure 20), suggesting that outliers form due to a sparse set of neurons across different ViTs.

**Highly activated neurons activate on all outlier locations.** Having seen that a small set of neurons highly activate on the top outlier, we now investigate whether these neurons highly activate across all outliers. We show three qualitative examples of activation maps in Figure 28 that closely align with the outliers. This alignment suggests that these highly activating neurons are responsible for outliers generally, which corroborates the results from OpenCLIP (Figure 3).

**Detecting register neurons.** Given that these highly activating neurons–which we refer to as "register neurons"–appear responsible for outlier formation, we develop an automatic discovery algorithm to identify and intervene upon them to shift outliers. To identify register neurons (Algorithm 1), we compute the average activations at outlier patches for all MLP neurons and return the top neurons.



Figure 28: **Highly activated neurons on the top outlier in DINOv2 activate on all outlier positions.** We present activation maps of three neurons from layer 17 that activate highly on the top outlier patch. These maps near-perfectly align with the high-norm outliers ("patch norms").

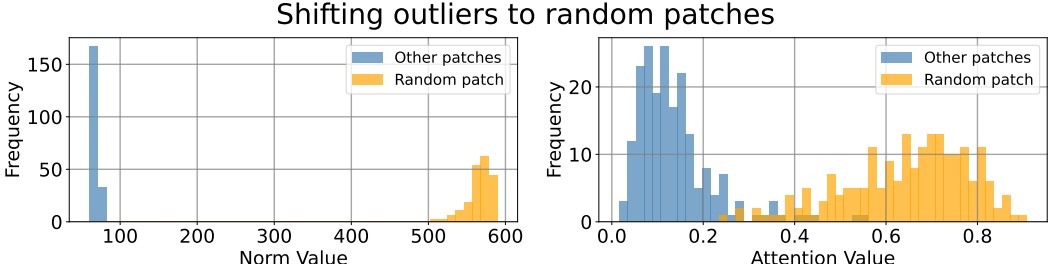

Figure 29: **Moving outliers in DINOv2 to random patches.** For all register neurons, we copy their highest activation into a selected patch and zero out the activations elsewhere. Left: norm of chosen random patch (yellow) and max norm of other patches (blue). Right: `[CLS]` attention to chosen random patch (yellow) and max `[CLS]` attention (blue) to any other patch. We find that the outliers shift to randomly selected patches, similar to OpenCLIP (Figure 4).

To move outliers to arbitrary patches, we follow the same intervention method from Section 3.2. For each register neuron, we copy the highest activation across patches into the selected patch and zero out the activations elsewhere. For applying Algorithm 1 to DINOv2, we set $top\_k = 45$, $highest\_layer = 17$, and the outlier threshold to 150.

**Register neurons causally set the position of outliers.** Following Section 3.2, we evaluate whether our intervention method can move outliers to arbitrary patches in DINOv2. After intervening, we measure the max norm of the selected patch, max norm of all other patches, and the highest last-layer `[CLS]` attentions to the selected patch and any other patch. We show in Figure 29 that our intervention successfully causes any selected patch to absorb the high norms and attention sinks. In Figure 1, we use register neurons to shift attention artifacts to form arbitrary spatial patterns, further demonstrating our ability to control outliers. These findings mirror that of OpenCLIP (Figure 4), exhibiting the generalizability of our intervention across ViTs.

## A.13 Extended Linear Probe Results

We present detailed linear probe results in Table 13 based on the experiments from Section 5.1. We run 3 independent seeds for our linear probe evaluations and report 95% confidence intervals below. Performance differences remain consistent. Models with test-time registers maintain or slightly improve performance across all metrics, with comparable performance to models trained explicitly with registers.

## A.14 Additional LOST object discovery results

**LOST Algorithm Overview.** The LOST unsupervised object discovery algorithm (Siméoni et al., 2021) begins by using patch representations to construct a Gram matrix $\mathcal{A}$, which encodes pairwise patch similarities. Next, we form an undirected patch similarity graph $\mathcal{G}$ where two nodes are connected if their similarity is positive. The initial seed is selected as the patch with the lowest degree in the graph. During the subsequent expansion phase, the algorithm iteratively selects the next lowest-degree patch that is positively correlated with the current seed, forming a set of patches $\mathcal{S}$.

| | IN Top-1 ↑ | ADE20k mIoU ↑ | NYUd rmse ↓ |
|---|---|---|---|
| DINOv2 ViT-L/14 | 86.36 ± 0.11 | 48.27 ± 0.15 | 0.3876 ± 0.0014 |
| w/ trained registers | 86.69 ± 0.09 | 49.07 ± 0.12 | 0.3823 ± 0.0011 |
| w/ test-time register | 86.38 ± 0.10 | 49.13 ± 0.11 | 0.3774 ± 0.0013 |
| OpenCLIP ViT-B/16 | 77.42 ± 0.08 | 40.14 ± 0.11 | 0.6025 ± 0.0017 |
| w/ test-time register | 77.53 ± 0.07 | 40.29 ± 0.09 | 0.5956 ± 0.0016 |

Table 13: **Linear probing results.** The performance of models with test-time registers maintains or improves performance over the unedited models, and largely matches models with trained registers.

A binary mask $\mathcal{M}$ is then computed by comparing all patch features to those in $\mathcal{S}$ and retaining the patches that, on average, have a positive correlation with features in $\mathcal{S}$. Finally, the bounding box of the connected component in $\mathcal{M}$ that contains the initial seed is used as the detected object.

**Visualizing Intermediate LOST Steps.** We visualize the impact of test-time registers on the intermediate computation steps of the LOST object discovery algorithm in Figure 30. The first row (LOST score) displays the inverse degree of each patch in the similarity graph $\mathcal{G}$. The second row shows the similarity between all patch features and the initial seed. The third row highlights the initial seed in yellow and illustrates the resulting seed expansion. For DINOv2, adding a test-time register cleans up the intermediate maps used during LOST, translating to the performance gains in Table 5. Notably, the resulting intermediate steps resemble those obtained when using trained registers. Adding a test-time register to OpenCLIP also refines the intermediate maps. However, since OpenCLIP already produces higher-quality intermediate LOST maps compared to DINOv2, the resulting improvement in unsupervised object discovery is marginal, which was also observed by Darcet et al. (2024).

**OpenCLIP LOST Feature Analysis.** We visualize the LOST score for OpenCLIP using the key, query, and value projection features in Figure 31. Adding a test-time register results in maps that are more focused on the object. However, the value projection from the original model already filters out much of the noise in the background regions, making the baseline maps relatively clean. As a result, the improvement from using a test-time register is less pronounced compared to DINOv2. Darcet et al. (2024) suggest that for OpenCLIP, this may be the case since outliers seem to reside in the null space of the value projection layer.

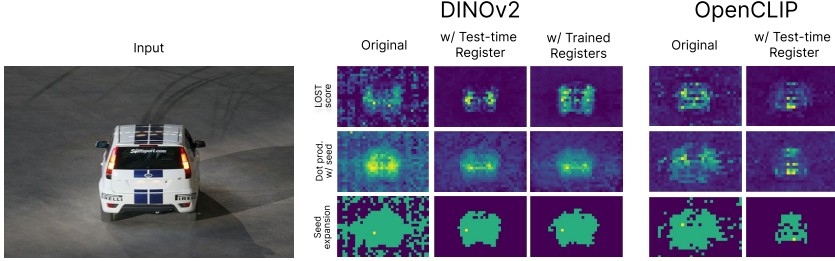

Figure 30: **Intermediate computation steps for LOST unsupervised object discovery.** Adding a test-time register produces sharper intermediate maps.

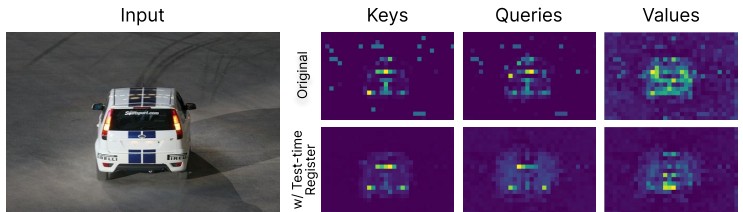

Figure 31: **OpenCLIP LOST maps using different features.** Adding a test-time register sharpens intermediate maps for different features. However, gains are limited for the values since the value projection already suppresses background noise.

### A.15 Additional VLM results

We present additional visualizations of LLaVA-Llama-3-8B patch norms and attention maps in Figure 32. As in Section 5.4, we visualize the patch norm map from the final layer of the vision encoder prior to projection into the language model input space. The patch norm maps highlight the presence of a sparse set of high-norm tokens, if any. We then visualize the average attention across all layers and heads of the language model from the response token to the visual tokens. We find that adding a test-time register to the vision encoder of the VLM leads to more interpretable attribution of the text output to the visual tokens.

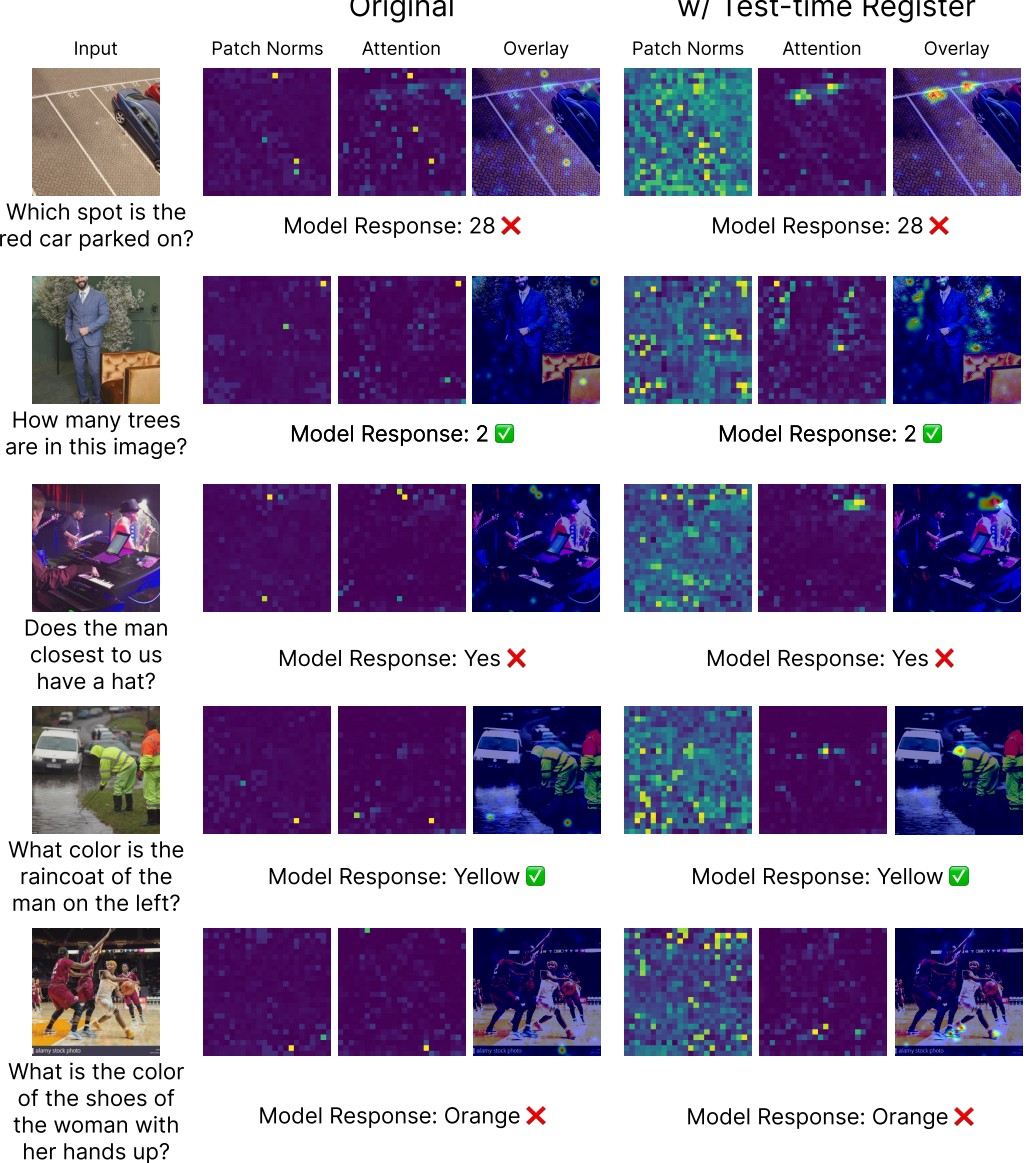

Figure 32: **Additional visualizations of LLaVA-Llama-3-8B patch norms and attention maps.** As in Figure 6, we show patch norms of the vision encoder and the average attention from the answer token to the visual tokens. Adding a test-time register mitigates high-norm artifacts in the vision encoder which would otherwise lead to anomalous attention patterns in the language model.

