# OpenReview forum: "Vision Transformers Don't Need Trained Registers"
_NeurIPS.cc/2025/Conference — NeurIPS 2025 spotlight_

### Official Review · Reviewer_SMDD · 2025-06-25

**Clarity:** 4
**Significance:** 3
**Originality:** 3
**Rating:** 5
**Confidence:** 4

**Summary:**

The authors introduce a method to move high-norm activations of "outlier tokens" that occur in Vision Transformers (ViTs) to register tokens. In contrast to previous methods dealing with this phenomenon, the authors' method does not require retraining the ViT. First, the authors identify the core mechanisms causing the occurrence of outlier tokens. Second, they use their findings to demonstrate that the mechanism can be employed to relocate outlier tokens to arbitrary positions within the image. Third, they utilize the mechanism to relocate outlier token activations to register tokens outside the image. The authors demonstrate that applying their method improves the model's performance on a range of downstream tasks.

**Questions:**

- How does the number of test-time register tokens influence the performance on the downstream task(s) (There is some information regarding multiple registers in Appendix A.1, but I didn't find metrics on downstream tasks for this ablation)?
- How does the experiment on typographic attacks support the author's claims?

**Ethical Concerns:**

["NO or VERY MINOR ethics concerns only"]

**Final Justification:**

The answers the authors provided were satisfactory.

**Limitations:**

Yes

**Quality:**

4

**Strengths And Weaknesses:**

**Strengths**

- The paper is well-written and leads the reader through the discovery process of the mechanism behind outlier tokens and how to move them to register tokens at test-time.
- The advancement of moving outlier tokens to register without re-training the model is meaningful.
- The authors conduct apt experiments.

**Weaknesses**

- The method performs on average slightly worse than adding registers by re-training the model.
- The pracitical relevance of the typographic attack mitigation using outlier tokens seems limited, because masking out the corresponding tokens in the input space works about equally well as moving high-norm tokens to the respective locations.

---

> ### Author Rebuttal · Authors · 2025-07-31
>
> We appreciate the reviewer’s thoughtful comments and questions. We are glad that they valued the discovery process of register neurons and found the proposed training-free solution meaningful. We address their questions regarding the number of test-time registers and the typographic attack mitigation below.
>
> **“How does the number of test-time register tokens influence the performance on the downstream task(s) (There is some information regarding multiple registers in Appendix A.1, but I didn't find metrics on downstream tasks for this ablation)?”**
>
> To quantify the effect of multiple test-time registers, we followed the procedure outlined in Appendix A.1 to create multiple test-time registers and then evaluated on unsupervised object discovery. We report the correct localization scores (corloc) below. Overall, we find that adding more test-time registers beyond one does not significantly change performance. We note that there is a slight degradation in performance beyond three test-time registers, although performance remains higher than without registers. We observe a similar trend with linear probe performance on NYUd v2 depth estimation below. These findings are consistent with the qualitative results in Figure 9: a single test-time register is sufficient to eliminate artifacts in the attention map, and additional registers have minimal impact.
>
> |              | VOC 2007 ↑ | VOC 2012 ↑ | COCO 20k ↑ |
> |--------------------------|------------|------------|------------|
> | DINOv2-L/14              | 32.2       | 36.8       | 25.4       |
> | + 1 test-time registers   | 53.8       | 57.9       | 41.9       |
> | + 2 test-time registers   | 53.9       | 57.8       | 41.7       |
> | + 3 test-time registers   | 52.8       | 57.0       | 40.6       |
> | + 4 test-time registers   | 51.9       | 55.6       | 39.3       |
> | + 5 test-time registers   | 50.5       | 54.4       | 38.2       |
>
> |            | NYUd RMSE ↓       |
> |--------------------------|-------------------|
> | DINOv2-L/14              | 0.3876 ± 0.0014    |
> | + 1 test-time registers   | 0.3774 ± 0.0013    |
> | + 2 test-time registers   | 0.3771 ± 0.0014    |
> | + 3 test-time registers   | 0.3785 ± 0.0014    |
> | + 4 test-time registers   | 0.3791 ± 0.0017    |
> | + 5 test-time registers   | 0.3795 ± 0.0016    |
>
>
> **“The practical relevance of the typographic attack mitigation using outlier tokens seems limited...How does the experiment on typographic attacks support the author's claims?”**
>
> Our intent with the typographic attack experiment is not to argue for the practical relevance of our mitigation strategy, but rather to demonstrate that the register neurons are highly causal to the model’s behavior (e.g., “only a small fraction of neurons to produce targeted changes in the image representation” in L246). In particular, this experiment (1) further supports our finding that editing the register neurons causally shifts the outlier patches and (2) demonstrates that local patch information at outliers is lost from the model’s computation. Darcet et al. (2024) showed that outlier patches retain less local information by demonstrating that their pixel/position information is harder to recover with a linear probe compared to normal patches (Section 2.1 of Darcet et al. (2024)). However, this analysis leaves open the possibility that local information at outliers might still be redistributed to other patches and indirectly influence the [CLS] token. Our results with the typographic attacks invalidate this possibility because shifting outliers on top of the text areas is performance-wise equivalent to the text never being present from the start (see pixel ablation baseline in the table below). The finding that the register neurons effectively mask patch information explains why mitigating the artifacts can improve dense prediction tasks.
>
> To further emphasize the highly causal nature of the register neurons, we applied a similar edit using three random sets of ten neurons and report the attack success rate with 95% confidence intervals below. Editing random neurons has minimal effect on the model’s output compared to modifying the ten register neurons. We will clarify these analysis points in the paper. However, we agree that the practical relevance of this method might be limited. We will consider swapping this section with the application of test-time registers on VLMs in Appendix A.9, which shows the practical value of adding registers post-hoc.
>
> | Method                 | Attack Success % ↓ |
> |------------------------|--------------------|
> | CLIP                   | 50.5               |
> | w/ Pixel Ablation      | 7.6                |
> | w/ register neuron edit| 7.5                |
> | w/ random neuron edit  | 50.5 ± 0.2   |

---

> > ### Comment · Reviewer_SMDD · 2025-08-01
> >
> > I thank the authors for the clarifications. My concerns have been adressed and I keep my high score.

---

### Official Review · Reviewer_YCz7 · 2025-06-28

**Clarity:** 3
**Significance:** 3
**Originality:** 3
**Rating:** 5
**Confidence:** 5

**Summary:**

The paper addresses the issue of high-norm outlier tokens in Vision Transformers (ViTs), which can lead to noisy attention maps. Prior solutions required retraining models with additional learned "register tokens." In contrast, the proposed method, test-time registers, offers a training-free alternative. It identifies a sparse set of "register neurons" responsible for generating high-norm activations and shifts these activations into additional, untrained tokens outside the image area. This redirection of outlier information results in cleaner attention maps, improved performance on downstream visual tasks, and increased robustness to typographic attacks.

**Questions:**

>  As no checkpoints with trained registers are available for OpenCLIP, we only compare our approach to the standard models.

1. Could you please clarify why you have not directly compared with the numbers reported in (Darcet et al. (2024))?

2. Please refer to the first and second points in **Weaknesses**.

**Ethical Concerns:**

["NO or VERY MINOR ethics concerns only"]

**Final Justification:**

My main concerns have been addressed. Therefore, I raise my score to 5.

**Limitations:**

Some of the limitations are discussed in Section 6.

**Quality:**

3

**Strengths And Weaknesses:**

**Strengths**

1. The proposed method is simple yet effective.
2. The paper is clearly written and easy to follow.
3. The effectiveness of the proposed method is analyzed across various downstream visual tasks (such as classification, dense prediction, zero-shot segmentation, and unsupervised object discovery).
4. The demonstrated use case showing how this approach can mitigate typographic attacks is also interesting and represents the usefulness of the proposed approach.

**Weaknesses**

1. Darcet et al. (2024) have shown that these outlier tokens are present in larger model sizes (Large, Huge, and Giant). In the experimental results (e.g. Table 2), the effectiveness of the proposed method is not analyzed across different model sizes.
2. In Table 5, the reported results for models with trained registers differ from those in Table 3 of  (Darcet et al. (2024)). Could you clarify the reason for this discrepancy? If the models used in their experiments differ from those in yours, why did you choose not to use the same models for a direct comparison?
3. In Table 4, I think "CLIP" should be replaced with "OpenCLIP" to avoid confusion.

---

> ### Author Rebuttal · Authors · 2025-07-31
>
> We appreciate the reviewer’s thoughtful comments and questions. We’re encouraged that the reviewer found our approach both simple and effective, and appreciated its clarity and breadth of evaluation.  We address comments regarding model scale and evaluation below.
>
> **“Darcet et al. (2024) have shown that these outlier tokens are present in larger model sizes... the effectiveness of the proposed method is not analyzed across different model sizes.”**
>
> Although Darcet et al. (2024) qualitatively show the presence of outliers in DINOv2-Giant, they perform quantitative downstream evaluation using ViT-B and ViT-L variants (Section 3.1: Training Algorithms and Data). Similarly, we evaluate using ViT-B and ViT-L variants. For instance, we use OpenCLIP ViT-B/16 and DINOv2-L/14 for linear probing evaluation as mentioned L181-186 in our manuscript. We will update the tables to explicitly note the model sizes used.
>
> We fully agree with you that evaluating at even larger scales is valuable. To that end, we have applied our method to InternViT-6B [1], one of the largest publicly available vision transformers. We applied our analysis from Section 3 and found that outliers form at layer 29 (out of 45). We then tracked the max norms across image patches at the output layer, and found that outlier patch norms reach up to a value of 7000, while the median patch norm value is 332 and the standard deviation is 396. Using Algorithm 1, we then identified 300 register neurons out of a total of 576,000 candidate neurons responsible for driving outlier formation. When we apply a test-time register with these register neurons, the maximum output patch norm at the output layer only reaches 608 on average, effectively mitigating the outliers. To assess the downstream impact, we ran unsupervised object discovery using the LOST algorithm (as in Section 5.3) since Darcet et al. (2024) found that the performance on this task correlates with the smoothness of a model’s attention maps. We report the correct localization (corloc) results below with and without a test-time register on InternViT-6B, and observe up to a 13-point correct localization improvement.
>
> | Model Variant          | VOC 2007 ↑ | VOC 2012 ↑ | COCO 20k ↑ |
> |------------------------|------------|------------|------------|
> | InternViT-6B           | 30.3       | 35.4       | 22.4       |
> | + Test-time register   | 41.6       | 48.7       | 32.7       |
>
> We will update the manuscript with this analysis and the corresponding visualizations showing outlier mitigation. We are particularly excited that this approach scales to larger models, suggesting it may also extend to large language models as well, where similarly high-norm outlier tokens have been observed [2].
>
> [1] Chen et al. “InternVL: Scaling up Vision Foundation Models and Aligning for Generic Visual-Linguistic Tasks.” CVPR 2024.
>
> [2] Dettmers et al. “LLM.int8(): 8-bit Matrix Multiplication for Transformers at Scale.” NeurIPS 2022.
>
> **“In Table 5, the reported results for models with trained registers differ from those in Table 3 of (Darcet et al. (2024)).” and “Could you please clarify why you have not directly compared with the numbers reported in (Darcet et al. (2024))?”**
>
> As noted in Section 3.1 of Darcet et al. (2024), their versions of OpenCLIP and DINOv2 were trained on different datasets than the original releases. For instance, their OpenCLIP was trained on a corpus based on Shutterstock while the original OpenCLIP is based on subsets of LAION-5b. Additionally, their checkpoints were not made publicly available. Instead, we used publicly released versions of the corresponding DINOv2 models provided by the DINOv2 team with and without registers, which were independently trained. We also use the original OpenCLIP checkpoints released by the OpenCLIP team. While this may introduce minor differences in performance with Darcet et al., it allows for a fair and reproducible evaluation using accessible models. We explain this in lines 181–186 of our manuscript, and are happy to further clarify this difference in the text if you see fit. We note that this situation also highlights a strength of our approach: it can be applied post-hoc to existing models, even when specialized register checkpoints are not available.
>
> **“In Table 4, I think "CLIP" should be replaced with "OpenCLIP" to avoid confusion.”**
>
> Thank you for catching that. We will update our next version with the fixed naming.

---

> > ### Comment · Reviewer_YCz7 · 2025-08-03
> >
> > I thank the authors for their clarifications and for providing additional results on a larger model. I believe it would be valuable to include these results in the camera-ready version. The rebuttal addressed my concerns.

---

> ### Author Response · Authors · 2025-08-04
>
> Thank you again for your thoughtful comments and for engaging with the discussion. We’re glad to hear that your concerns were addressed. Since those concerns were central to your earlier assessment, we’d appreciate it if you might consider updating your score to reflect your current impression, if you are open to it. If any questions remain, we’re happy to follow up further. We appreciate your time and engagement with the paper.

---

> > ### Comment · Reviewer_YCz7 · 2025-08-06
> >
> > Having found the rebuttal satisfactory, I have increased my score.

---

### Official Review · Reviewer_fpZR · 2025-07-03

**Clarity:** 3
**Significance:** 3
**Originality:** 2
**Rating:** 5
**Confidence:** 1

**Summary:**

This work investigates the emergence of high-norm tokens caused by a sparse set of neurons focusing strong activations on outlier tokens. This phenomenon, observed in models like CLIP and DINOv2, degrades downstream performance. While prior solutions require retraining with learned register tokens to suppress these effects, the authors propose a training-free alternative: redirecting high-norm activations into a new, untrained token during inference. This method cleans attention maps, improves task performance, and matches the effectiveness of retrained models with register tokens, offering a practical fix for existing pre-trained models.

**Questions:**

Can you develop a theoretical framework based on simple models or simple arguments for register neurons?

**Ethical Concerns:**

["NO or VERY MINOR ethics concerns only"]

**Final Justification:**

The reply from the authors is satisfactory and introduce a new interesting perspective. I therefore recommend the paper to be accepted (my vote would be between 4 and 5).

**Limitations:**

yes

**Quality:**

3

**Strengths And Weaknesses:**

Strengths: a thorough analysis of the mechanism behind the creation of high-norm tokens in ViT, proposing a training-free method to solve the problem, showing that it is comparable in quality to alternatives based on retraining.

Weaknesses: it would have been interesting to have more theory on "register neurons" either from simple models, or simple arguments.

---

> ### Author Rebuttal · Authors · 2025-07-31
>
> We appreciate the reviewer’s thoughtful comments. Several papers have investigated the emergence of outlier tokens, particularly in large language models,  and have proposed various broad explanations for why they arise. One explanation is the so-called "no-op" (no-operation) hypothesis [1,2,3], which suggests that the Softmax constraint — requiring attention weights to sum to one — prevents the model from assigning zero weight to all tokens. As a result, the attention layer is forced to aggregate *some* information from other tokens, even when the current embeddings already contain sufficient information for the task. This often leads the model to allocate excess attention to specific, low-information tokens, a phenomenon known as "attention sink" [4]. Further empirical work suggests that these outlier tokens emerge to introduce an implicit, input-independent bias term into the attention mechanism [5,6]. While our submission aims to provide a mechanistic analysis of this phenomenon in ViTs along with a practical solution, in this response, we provide a simple mathematical model. Specifically, we illustrate how under certain conditions, "register neurons" can induce high-norm tokens that attract excess attention under the "no-op" hypothesis.
>
> ## Setup
> We consider the MLP layer from the penultimate Transformer block followed by the full final block (attention + MLP) of a residual Vision Transformer without normalization layers. Without loss of generality, let the input sequence to the penultimate MLP be a task-specific CLS token and two patch tokens: $x^{(0)}_{cls}, x^{(0)}_1, x^{(0)}_2\in \mathbb{R}^{d}$. Stacking them vertically will form our input $X\in\mathbb{R}^{3\times d}$.
>
> The MLPs will consist of two linear layers (no bias) and an activation function $\phi$ (e.g., ReLU or GELU). We denote the weight matrices of the penultimate block's MLP as $W_1^{(1)} \in \mathbb{R}^{d \times d_{\text{mlp}}}, W_2^{(1)} \in \mathbb{R}^{d_{\text{mlp}} \times d}$, and those of the final block's MLP as $W_1^{(2)} \in \mathbb{R}^{d \times d_{\text{mlp}}}, W_2^{(2)} \in \mathbb{R}^{d_{\text{mlp}} \times d}$.
>
> There will be one attention head with identity projection matrices: $W_Q = W_K = W_V = W_O = I_d \in \mathbb{R}^{d \times d}$. Finally, the output task-specific head is a linear projection that uses the [CLS] token's embedding for prediction: $W_{\text{h}} \in \mathbb{R}^{d \times c}$,  where $c$ is the number of output classes (or regression targets).
>
> ## Analysis
> Under the "no-op" hypothesis, the $x_{cls}$ embedding before attention has sufficient information for the task and does not need mixing with other tokens via attention. We will show how a "register neuron" can induce outlier behavior for token $x_2$.
>
> Let $(W\_1^{(1)})_{:, 1}$ = $u\_1 \in ker(W_1^{(2)\top}) \\cap ker(W^\top_h)$ be a "register neuron" and the corresponding column in the second MLP matrix be $(W\_2^{(1)})\_{1, :}$  = $u\_2$ $\in ker(W\_1^{(2)\top}) \\cap ker(W^\top\_h)$. After the first MLP with a residual connection, the updated hidden states for the tokens are:
>
> 1\) $x^{(1)}\_{cls}$ = $x^{(0)}\_{cls}+\phi(x^{(0)}\_{cls}W_1^{(1)})W_2^{(1)}$
>
> 2\) $x^{(1)}\_{1}$ = $x^{(0)}\_{1}+\phi(x^{(0)}\_{1}W_1^{(1)})W_2^{(1)}$
>
> 3\) $x^{(1)}\_{2}$ = $x^{(0)}\_{2}+\phi(x^{(0)}\_{2}W_1^{(1)})W_2^{(1)}$
>
> This can be interpreted as adding a vector in $\mathbb{R}^d$ to the tokens modulated by some coefficient $\alpha$:
>
> 4\) $    x^{(1)}\_{cls} = x^{(0)}\_{cls}+\alpha_1v\_1$
>
> 5\) $    x^{(1)}\_{1} = x^{(0)}\_{1}+\alpha_2v\_2$
>
> 6\) $    x^{(1)}\_{2} = x^{(0)}\_{2}+\alpha_3v\_3$
>
> If $x^{(0)}\_2$ is strongly aligned with the register neuron $u_1$ (i.e., $x^{(0)}_2 = \beta u_1$ for some large, positive constant $\beta$), its activation magnitude $\tilde{\alpha}\_3 = \phi(x_2^{(0)\top} u_1)$ will be large and the corresponding column in the second MLP matrix $u_2$ will be the dominant direction added to $x_2$ in the residual stream. Thus, $ x^{(1)}\_{2} \approx x^{(0)}\_{2}+\alpha_3u\_2 = \beta u\_1+\alpha\_3u\_2$, where $\alpha\_3\approx\tilde{\alpha}\_3 \gg \alpha\_1, \alpha\_2$ and thus $\|x^{(1)}\_{2}\| \gg \|x^{(1)}\_{cls}\|, \|x^{(1)}\_{1}\|$.
>
> Given identity projection matrices during attention (i.e., the key, queries, values are the tokens themselves), the update to the [CLS] token after attention is $x^{(2)}\_{cls} = x^{(1)}\_{cls} + p\_{cls}x^{(1)}\_{cls}+ p\_1x^{(1)}\_{1} + p\_2x^{(1)}\_{2}$, where $[p\_{cls}, p\_1, p\_2] = \text{Softmax}([x^{(1)\top}\_{cls}x^{(1)}\_{cls}, x^{(1)\top}\_{cls}x^{(1)}\_{1}, x^{(1)\top}\_{cls}x^{(1)}\_{2} ])$.
>
> As $ \|x^{(1)}\_{2}\| \to \infty $, the dot product $x^{(1)\top}\_{cls} x^{(1)}\_{2} = \|x^{(1)}\_{cls}\| \cdot \|x^{(1)}\_{2}\| \cdot \cos(\theta)$ grows without bound even for small angle $\theta$ between the two token embeddings, leading the Softmax to assign nearly all attention weight to $x^{(1)}_{2}$. It follows that:
>
> 7\) $x^{(2)}\_{cls} \approx x^{(1)}\_{cls} + x^{(1)}\_{2} = x^{(1)}\_{cls} + \beta u\_1+\alpha_3u\_2 $.
>
> Since, $u\_1, u\_2 \in ker(W\_1^{(2)\top})$, the update to the [CLS] token after the final block's MLP is
>
> 8\) $x^{(3)}\_{cls} = x^{(2)}\_{cls}+\phi(x^{(2)}\_{cls}W\_1^{(2)})W\_2^{(2)} = x^{(1)}\_{cls} + \beta u\_1+\alpha\_3u\_2 +\phi(x^{(1)}\_{cls}W\_1^{(2)})W\_2^{(2)}$
>
> Finally, the output prediction is:
> 9\) $x^{(3)}\_{cls}W\_{h} = (x^{(1)}\_{cls} + \beta u\_1+\alpha\_3u\_2 +\phi(x^{(1)}\_{cls}W\_1^{(2)})W\_2^{(2)})W\_{h}$.
>
> As $u\_1, u\_2 \in ker(W^\top\_{h})$, the output will be:
>
> 10\) $(x^{(1)}\_{cls} +\phi(x^{(1)}\_{cls}W_1^{(2)})W\_2^{(2)})W\_{h}$
>
> which is equivalent to the forward pass without the attention layer, thus achieving the "no-op" attention behavior. This formalism shows how a "register neuron" can induce a high norm token, an "attention sink," and result in a null attention update in line with the "no-op" hypothesis.
>
> ## Empirical Connections
>
> ### (1) Task-Irrelevant Attention Sinks:
>
> The token that achieves the attention sink behavior in this model lives in the null space of the task specific head, suggesting it is not informative of the task. This aligns with the observation that register neurons fire on seemingly random, uninformative regions such as background or uniform texture.
>
> ### (2) Lack of Attention Sink Ruins Performance:
> Secondly, this model suggests that zeroing out the activation of a register neuron would disrupt the "attention sink" and "no-op" attention behavior. Thus, the attention would unnecessarily alter the [CLS] token—which was already sufficiently informative prior to attention—and degrade its representation. To test this, we zero-out the 10 register neurons in OpenCLIP ViT-B/16 and observe a drop in IN1k zeroshot performance from 70.2% to 55.6%. As a baseline, zeroing out a random set of 10 neurons over three trials results in minimal change: 69.3% ± 1.1.
>
> ### (3) Register Neurons Induce an Implicit Attention Bias:
>
> We empirically observe that register neurons exhibit a consistently high activation value regardless of input (see Fig. 3 of paper). In our mathematical model, the attention update for the [CLS] token (equation 7) thus reads in a constant high norm vector which serves as an additive bias term during the attention operation, in line with observations from [5,6]. Sun et al. (2024) [5] proposed learning bias terms in the attention layers of large language models to remove outlier tokens. Specifically, given the query, key, and value matrices $Q, K, V \in \mathbb{R}^{T\times d}$ for each attention head, they train additional parameters $\mathbf{k}', \mathbf{v}' \in \mathbb{R}^{d}$ such that
>
> $$
> \text{Attention}(Q, K, V; \mathbf{k}', \mathbf{v}') =
> \text{softmax}\left(
> \frac{Q \begin{bmatrix} K^T & \mathbf{k}' \end{bmatrix}}{\sqrt{d}}
> \right)
> \left[
> \begin{array}{c}
> V ;\\
> \mathbf{v}'^T
> \end{array}
> \right]
> $$
>
> To test that the register neurons implicitly induce attention biases, we use OpenCLIP ViT-B/16 and set $\mathbf{k'}$ and $\mathbf{v'}$ for each attention head to the mean key and value vectors of a test-time register token holding register neuron activations, averaged across 1000 images. At inference time, we disable this test-time register token and instead inject $\mathbf{k}'$ and $\mathbf{v}'$ directly into the attention computation as above (we still zero out the register neurons). This maintains the $70.2\\%$ IN1k zero-shot performance while reducing the average maximum output patch norm from $150$ down to $40$. The median patch norm is 25, indicating that outlier activations are effectively suppressed.
>
>
> We have shown that this simple model demonstrates how register neurons can induce high-norm tokens and attention sinks in a simple setting, aligning with our empirical observations. If you find that this perspective is insightful, we can update the paper to include this analysis.
>
>
> [1] Clark et al. "What does bert look at? an analysis of bert’s attention." ACL Workshops 2019.
>
> [2] Kobayashi et al. "Attention is Not Only a Weight: Analyzing Transformers with Vector Norms" EMNLP 2020.
>
> [3] Bondarenko et al. "Quantizable transformers: Removing outliers by helping attention heads do nothing." NeurIPS 2023.
>
> [4] Xiao et al. "Efficient Streaming Language Models with Attention Sinks." ICLR 2024.
>
> [5] Sun et al. "Massive activations in large language models." COLM 2024.
>
> [6] Gu et al. "When attention sink emerges in language models: An empirical view." ICLR 2025.

---

> > ### Comment · Reviewer_fpZR · 2025-08-04
> >
> > Thanks for the reply which I find satisfactory. This perspective is insightful, please update the paper to include this analysis. I have increased my rating in consequence.

---

### Official Review · Reviewer_xEEL · 2025-07-03

**Clarity:** 3
**Significance:** 4
**Originality:** 4
**Rating:** 6
**Confidence:** 4

**Summary:**

This work follows previous works that introduced register tokens when training vision transformers to prevent the appearance of high norm tokens artifacts during inference. While register tokens are effective, they need to be trained alongside a model, making them impractical to include for already trained models.
In this work, the authors find that a small set of specific neurons can be identified that consistently cause these high activations over various images. Their activations on patches with artifacts can be shifted to a register added at test-time to fix the artifacts while preserving their global information. This approach allows to address the same issue as trained register tokens, but without the need for any training. The results show that these test-time registers maintain or improve performance on various tasks. The improvements are particularly strong on tasks that rely on having smooth feature maps, such as unsupervised object discovery, although it does not reach the level of trained register tokens.

**Questions:**

1. Can the authors define earlier what they mean by "neurons"? Understanding what a neuron consists of is key to this work, but a brief technical definition only appears in the related works (line 79). While this definition is complete, I think it is really critical to ensure that the reader grasps and visualises exactly what is meant by "neuron" as early as possible, potentially using a more extensive explanation. One potential way to do so would be to visualise attention maps stacked from different neurons.
2. In the checklist, the authors claim that the full algorithm is described in Algorithm 1. However, this only details the process to identify register neurons, not the steps to move them to test-time registers. Can the authors please include a full description of the algorithm that includes register token creation at test time (see my comment on clarity)?
3. Could the authors please clarify how Figure 3 (right) highlights that the small subset of <10 neurons are consistent across images? The fact that the neurons are consistent across images is very valuable, but I am currently failing to understand how that is visible from the results shown in the paper.
4. Do the authors have any insights or results on why it is useful to shift register neuron activations to a new token at test-time, and if the model does indeed rely on attention to the added register to make predictions? It would seem like trained models could have learned to rely on positional encodings to affect the attention process, including to use the global information stored in register neurons. Is there a reason that we can expect the model to still be able to use the global information from the register neurons, even after shifting their activations to the register token? One additional test to validate that the model does use the test-time register could be to remove activations from register neurons (as is done when shifting them to the register token), but not include the register token.
5. It would appear to me that the most critical aspect of preventing typographic attacks (Section 5.4) might be the modification of activations located in patches corresponding to text. Could the authors discuss more if it is critical that register neurons’ activations are used for this rather than simply setting zeros or random values in the same activations?

**Ethical Concerns:**

["NO or VERY MINOR ethics concerns only"]

**Final Justification:**

The authors have addressed all my concerns in the rebuttal and have included new experiments that further validate their approach.

Considering the novel findings of this work (the existence of register neurons), the comprehensive proposition and validation of a practical algorithm to manage them (test-time registers), and the significant impact this can have (improving off-the-shelf transformers across diverse tasks without requiring expensive re-training), I recommend this paper for acceptance.

**Limitations:**

I find the limitations discussed in Section 6 sufficient.

**Paper Formatting Concerns:**

I have no major concerns about formatting.

**Quality:**

3

**Strengths And Weaknesses:**

(+) = strength, (-) = weakness

**Quality**:
1. (+) The paper is well motivated and proposes a new solution to a real observed known issue.
2. (+) The study of the mechanisms leading to the observed artifacts produces multiple valuable insights, leading to the design of a good solution.
3. (+) The performance of the proposed test-time register token that gathers activations from register neurons is good, improving or maintaining performance of models. The value of this approach is strongly validated by all the results.
4. (+) The design choices (number of register tokens, initialisation) are motivated with experiments in the appendix.
5. (+) Limitations are clearly acknowledged and discussed.
6. (-) The results do not seem to be from multiple runs for statistical significance (based on the Checklist answer 7), even though linear probes are trained. The consistency would be valuable to validate considering how close some of the results are (e.g., Table 2).
7. (-) The approach requires setting some hyperparameters, and it is unclear how involved this might be for new models.
8. (-) The paper does not discuss why we would expect that shifting activations to a new token would still be valuable to the model without any re-training. While the experiments validate that the test-time register keeps global information, they do not validate that it still has a role in the model's predictions, and that improvements in the results come from more than disabling the artifacts from register neurons.

**Clarity**:
1. (+) The paper is well structured. I find the flow to support a good understanding of the work, with a strong study of the mechanisms before introducing the proposed solution and then the results.
2. (-) I find that the full practical approach would benefit from being described in more detail and in one place, either through an algorithm, figure, or mathematical formulation. Algorithm 1 helps make the register neuron identification process very clear, but I believe that how we get to a register token at test time is not as clear due to the information to understand the whole process being split over multiple sections. I believe that it would be helpful to have a clear location in the paper that would make this whole process very clear, so that the reader clearly understands that register neurons are fixed for new images, but outlier patch locations have to be identified, etc.

**Significance**:
1. (+) This paper has strong significance. It proposes a method that can potentially improve any off-the-shelf vision transformers without retraining, and presents valuable new insights about the functioning of the models studied.

**Originality**:
1. (+) The proposed solutions are grounded in original findings about the behaviour of specific neurons and lead to a novel algorithm to manage them, making this work original.

---

> ### Author Rebuttal · Authors · 2025-07-31
>
> We appreciate the reviewer’s thoughtful comments.  We are glad that they found the submission to contain valuable insights with strong significance. We address their comments below.
>
> **“Can the authors define earlier what they mean by ‘neurons’?...the reader grasps and visualises exactly what is meant by ‘neuron’ as early as possible”**
>
> Great point. We will move the formal definition — post-nonlinearity, single-channel activations — earlier into the introduction. Specifically, given a token representation $ x_i \in \mathbb{R}^{d_{\text{model}}}$, we define neurons as the individual scalar activations along each channel dimension after applying the first linear projection and nonlinearity in the MLP, i.e., $ \phi(x_i W_{\text{in}} + b_{\text{in}}) $, where $ \phi $ is the activation function. The individual channels for each patch token can be reshaped into a spatial grid to produce a visualization such as that in Fig. 4. We will also add a visual example of neuron-wise activation maps (e.g., stack maps similar to those in Fig. 4) as suggested.
>
> **“Can the authors please include a full description of the algorithm that includes register token creation at test time?”** and **“have a clear location in the paper that would make this whole process very clear.”**
>
> Adding a test-time register involves “initializ[ing] our extra token to be a vector of zeros” (L154) and then whenever a register neuron is encountered during the forward pass, “copy[ing] the highest [register] neuron activation across the tokens” (L133-134) into the test-time register and then “zero[ing] out the activations of the [register] neuron elsewhere” (L134-135). Here is the pseudocode for shifting the register neurons in each layer:
> ```
> Register_neurons: List[int], register neuron indices in this layer
> Activation_maps: List[# tokens, # MLP neurons], neuron activation maps produced by this layer
> SET_TT_REGISTER(register_neurons, activation_maps):
>     For neuron in register_neurons:
>         # set test-time register token (idx = -1) to max register neuron activation value over all tokens
> 	Activation_maps[-1, neuron] = max(Activation_maps[:, neuron])
>         # zero-out register neuron activation for all other tokens
> 	Activation_maps[:-1, neuron] = 0
>     Return Activation_maps
> # proceed with forward pass
>  ```
> We will update Section 4 with the above pseudocode. We have also created a figure that overlays neuron activation maps on the ViT architecture, illustrating how register neurons are identified, fixed across new images, and how their activations are shifted into the test-time register. We will incorporate this figure to consolidate the full test-time register procedure.
>
> **“The results do not seem to be from multiple runs for statistical significance… e.g., Table 2).”**
>
> We have now run 3 independent seeds for our linear probe evaluations from Table 2 and report 95% confidence intervals below. Performance differences remain consistent. We will update Table 2 in the paper.
>
> | Model Variant            | IN Top1 ↑        | ADE20k mIoU ↑     | NYUd rmse ↓       |
> |--------------------------|------------------|-------------------|-------------------|
> | DINOv2 ViT-L/14          | 86.36 ± 0.11     | 48.27 ± 0.15      | 0.3876 ± 0.0014   |
> | + trained registers      | 86.69 ± 0.09     | 49.07 ± 0.12      | 0.3823 ± 0.0011   |
> | + test-time register     | 86.38 ± 0.10     | 49.13 ± 0.11      | 0.3774 ± 0.0013   |
> | OpenCLIP ViT-B/16        | 77.42 ± 0.08     | 40.14 ± 0.11      | 0.6025 ± 0.0017   |
> | + test-time register     | 77.53 ± 0.07     | 40.29 ± 0.09      | 0.5956 ± 0.0016   |
>
> **“The approach requires setting some hyperparameters, and it is unclear how involved this might be for new models.”**
>
> Our approach relies on three hyperparameters: (1) the outlier threshold, (2) the index of the highest layer we search up to when identifying register neurons, and (3) the number of register neurons to return. The first two are straightforward to set using simple heuristics. Specifically, the outlier threshold can be defined using the classical criterion of three standard deviations above the mean patch norm. The second, *top_layer*, is chosen as the layer at which outliers first emerge. Setting the number of neurons to return is slightly more involved, relying on sweeping *top_k* until the outliers are suppressed. However, this is manageable in practice since the number of register neurons is sparse.
>
> To test how transferable the approach is, we have applied our method to InternViT-6B [1], one of the largest publicly available vision transformers. We applied our analysis from Section 3 and found that outliers form at layer 29 (out of 45). We then tracked the max norms across image patches at the output layer, and found that outlier patch norms reach up to a value of 7000, while the median patch norm value is 332 and the standard deviation is 396. Using Algorithm 1, we then identified 300 register neurons (out of 576,000) responsible for driving outlier formation. When we apply a test-time register, the maximum output patch norm at the output layer reaches only 608 on average, mitigating the outliers. To assess downstream impact, we ran unsupervised object discovery (as in Section 5.3) and report correct localization (corloc) results below. We observe up to a 13-point corloc improvement.
>
> |          | VOC 2007 ↑ | VOC 2012 ↑ | COCO 20k ↑ |
> |------------------------|------------|------------|------------|
> | InternViT-6B           | 30.3       | 35.4       | 22.4       |
> | + Test-time register   | 41.6       | 48.7       | 32.7       |
>
>
> We will update the manuscript to include this analysis, corresponding visualizations, and the discussion of hyperparameter selection from above. We are particularly excited that this approach scales to larger models, suggesting it might extend to large language models as well, where similarly high-norm outlier tokens have been observed [2].
>
> [1] Chen et al. “InternVL: Scaling up Vision Foundation Models and Aligning for Generic Visual-Linguistic Tasks.” CVPR 2024.
>
> [2] Dettmers et al. “LLM.int8(): 8-bit Matrix Multiplication for Transformers at Scale.” NeurIPS 2022.
>
>
>
> **"Could the authors please clarify how Figure 3 (right) highlights that the small subset of <10 neurons are consistent across images?"**
>
> The distribution in Figure 3 (right) shows the average activations across 1000 images of individual neurons on the top outlier patch. There are <10 neurons with extremely high average activations, suggesting that these neurons are on-average highly activated on the top outlier patch. As most of the average activations are close to zero, we opted to only take the top 1% of neurons to highlight the neurons with high average activations. We will include more activation maps (formatted like Figure 4) from the same neurons to further clarify that the set of register neurons is consistent across images.
>
> **“...why is it useful to shift register neuron activations to a new token at test-time, and if the model does indeed rely on attention to the added register to make predictions?”**
>
> To verify that the model uses the added register, we experimented with removing the high norm activations from the register neurons (setting them to zero) without including the test-time register, as suggested.  On OpenCLIP ViT-B/16, zero-shot ImageNet performance drops from 70.2% to 55.6%, while adding the test-time register maintains the performance of 70.2%. As a baseline, zeroing out a random set of 10 neurons over three trials results in minimal change: 69.3%±1.1.  We then assessed attention patterns by computing the average last-layer attention from the [CLS] token to the test-time register across 1000 ImageNet images. The [CLS] token attends to the test-time register with an average score of 0.1657, while the median of the attention to all other tokens is only 0.0006.  Combined with Table 1, which shows that test-time registers hold global information, this strongly suggests that the model relies on the register during the internal computation. The [CLS] token actively reads from it, integrating its global information via attention. We will add this discussion to Section 3 to motivate why we must shift the register neuron activations rather than deleting them.
>
>
> **“… is it critical that register neurons’ activations are used for this rather than simply setting zeros or random values in the same activations?”**
>
>
> We augmented the typographic attack results (Table 6) with the suggested baselines and present them below. First, we zeroed out the register neuron activations (w/ register neurons zeroed).  Second, we replaced the register neuron activations corresponding to the text area with values randomly drawn from a normal distribution matched to their activation mean and variance (w/ random register neuron value). Third, we applied the same editing strategy – shifting the max activations – but did so on a random set of ten neurons instead of the ten identified register neurons (w/ random neuron edit). For all stochastic baselines, we report the 95% confidence intervals over three runs below.  We find that using both the register neurons and their specific outlier activations are critical for masking the adversarial text in the model’s internal representations. This highlights that register neuron outlier activations are highly causal and editing them is performance-wise equivalent to the text never being present from the start (see pixel ablation baseline below).  We will update the table in the paper.
>
> |                    | Attack success % ↓ |
> |------------------------------|---------------------|
> | CLIP                          | 50.5                |
> | w/ Pixel Ablation             | 7.6                 |
> | w/ register neuron edit       | 7.5                 |
> | w/ register neuron zeroed     | 48.4                |
> | w/ random register neuron value | 48.2 ± 1.5        |
> | w/ random neuron edit         | 50.5 ± 0.2          |

---

> > ### Comment · Reviewer_xEEL · 2025-08-04
> >
> > I thank the authors for their careful consideration of my comments.
> >
> > They have very clearly addressed all my concerns and questions, and have provided strong additional results to support their responses.
> >
> > I believe that the proposed updates to the paper will further improve its already strong clarity and validation of the proposed approach.

---

> > > ### Author Response · Authors · 2025-08-07
> > >
> > > Thank you for your engagement throughout the process. We’re glad the rebuttal fully addressed your concerns and that you found the additional results strong. We will update the paper with these revisions.

---

> ### Author Response · Authors · 2025-08-04
>
> Thank you again for your thoughtful review. We’ve done our best to address your points, and your feedback has improved the paper. With two days left in the discussion period, we’d be grateful if you’re able to take a final look. Since your concerns were mainly about clarity, we hope the revisions help. If the changes better reflect your view of the work’s contribution, we’d appreciate your consideration in updating your score, if you're open to it. We’re happy to clarify anything further.

---

### Note · Authors · 2025-08-12

We would like to thank the reviewers for their constructive feedback and thoughtful discussion. Reviewer xEEL described our proposed solution as “novel” and “grounded in original findings,” with “strong significance” and the potential to “improve any off-the-shelf vision transformer without retraining.” Reviewers appreciated the mechanistic analysis, stating that our work provides “a strong study of the mechanisms” (xEEL), “a thorough analysis of the mechanism behind the creation of high-norm tokens in ViT” (fpZR), and “leads the reader through the discovery process of the mechanism behind outlier tokens” (SMDD). They highlighted the practical significance of our approach, describing it as “simple yet effective” with “improved performance on downstream visual tasks” (YCz7) and noting that “the advancement of moving outlier tokens to registers without re-training the model is meaningful” (SMDD).

All reviewers indicated that their concerns have been addressed, and we will incorporate the promised changes. Notably, in response to reviewers xEEL and YCz7, we have applied our method to InternViT-6B [1], one of the largest publicly available Vision Transformers, and observed up to 13-point correct localization improvement in unsupervised object discovery. We are particularly excited that this approach scales up to larger models, suggesting it may also extend to large language models as well, where similar high-norm outlier tokens have been observed [2]. In response to reviewer fpZR, we will add our mathematical model of register neurons into the paper. In response to SMDD, we will swap our typographic attacks experiment with the application of test-time registers on Vision-Language Models from Appendix A.9, highlighting the broader applicability of test-time registers.

[1] Chen et al. “InternVL: Scaling up Vision Foundation Models and Aligning for Generic Visual-Linguistic Tasks.” CVPR 2024.

[2] Dettmers et al. “LLM.int8(): 8-bit Matrix Multiplication for Transformers at Scale.” NeurIPS 2022.

---

### Decision · Program_Chairs · 2025-09-17

**Decision:**

Accept (spotlight)

**Comment:**

This paper proposes a training-free approach to registers for Vision Transformers. It has received all positive reviews.

The paper was found well written with the method being clear, though more explanations were recommended (wxZe, Ffhe). The method was found new (29Ti, wxZe, ykSW), well motivated (29Ti, Ffhe), with good qualities (29Ti, wxZe), accompanied by theory that may be useful elsewhere (wxZe, ykSW). The experiments were found well designed and comprehensive (ykSW, Ffhe), though additional experiments were recommended (wxZe, Ffhe) as well as clarifications on the setup (Ffhe).

The paper was found well structured (xEEL), well written (YCz7, SMDD). The method was found well motivated (xEEL), insightful (xEEL, SMDD), significant (xEEL), simple (YCz7), including thorough analysis (fpZR). The method was found effective (xEEL, fpZR, YCz7), on appropriate experiments (YCz7, SMDD). The design choices were supported by experiments (xEEL). The limitations were found adequately discussed (xEEL).

Concerns included the need for more detail (xEEL), statistical significance (xEEL), hyperparameters (xEEL), interpreting why the model works (xEEL), lack of theory (fpZR), lack of different model sizes (YCz7), inconsistencies with prior work (YCz7), inferior performance relative to training (SMDD).

The authors provided detailed feedback in their rebuttal, accompanied with new experiments where needed. This feedback addressed all reviewers' concerns. The authors committed to update the paper according to the new material they have presented during the discussion.

Apart from the high ratings, the work was found of strong significance, strong evaluation and results by at least one reviewer. Spotlight presentation is therefore recommended.